# Indexing and partitioning the spatial linear model for large data sets

**Jay M. Ver Hoef** [1]*, **Michael Dumelle** [2], **Matt Higham** [3], **Erin E. Peterson** [4], **Daniel J. Isaak** [5]

**1** Marine Mammal Laboratory, NOAA-NMFS Alaska Fisheries Science Center, Seattle, WA, United States of America, **2** United States Environmental Protection Agency, Corvallis, Oregon, United States of America, **3** St. Lawrence University Department of Mathematics, Computer Science, and Statistics, Canton, New York, United States of America, **4** Australian Research Council Centre of Excellence in Mathematical and Statistical Frontiers (ACEMS), Queensland University of Technology, Brisbane, Queensland, Australia, **5** Rocky Mountain Research Station, U.S. Forest Service, Boise, ID, United States of America

☯ These authors contributed equally to this work.

* jay.verhoef@noaa.gov

**Data Availability Statement:** The SPIN method has been implemented in the spmodel R package https://cran.r-project.org/web/packages/spmodel/index.html. The example data can be downloaded

## Abstract

We consider four main goals when fitting spatial linear models: 1) estimating covariance parameters, 2) estimating fixed effects, 3) kriging (making point predictions), and 4) block-kriging (predicting the average value over a region). Each of these goals can present different challenges when analyzing large spatial data sets. Current research uses a variety of methods, including spatial basis functions (reduced rank), covariance tapering, etc, to achieve these goals. However, spatial indexing, which is very similar to composite likelihood, offers some advantages. We develop a simple framework for all four goals listed above by using indexing to create a block covariance structure and nearest-neighbor predictions while maintaining a coherent linear model. We show exact inference for fixed effects under this block covariance construction. Spatial indexing is very fast, and simulations are used to validate methods and compare to another popular method. We study various sample designs for indexing and our simulations showed that indexing leading to spatially compact partitions are best over a range of sample sizes, autocorrelation values, and generating processes. Partitions can be kept small, on the order of 50 samples per partition. We use nearest-neighbors for kriging and block kriging, finding that 50 nearest-neighbors is sufficient. In all cases, confidence intervals for fixed effects, and prediction intervals for (block) kriging, have appropriate coverage. Some advantages of spatial indexing are that it is available for any valid covariance matrix, can take advantage of parallel computing, and easily extends to non-Euclidean topologies, such as stream networks. We use stream networks to show how spatial indexing can achieve all four goals, listed above, for very large data sets, in a matter of minutes, rather than days, for an example data set.

from the Github repository, https://github.com/jayverhoef/midColumbiaLSN.git.

**Funding:** JVH: The project received financial support through Interagency Agreement DW-13-92434601-0 from the U.S. Environmental Protection Agency (EPA), and through Interagency Agreement 81603 from the Bonneville Power Administration (BPA), with the National Marine Fisheries Service, NOAA. The funders had no role in study design, data collection and analysis, decision to publish, or preparation of the manuscript.

**Competing interests:** The authors have declared that no competing interests exist.

## Introduction

The general linear model, including regression and analysis of variance (ANOVA), is still a mainstay in statistics,

$$\mathbf{Y} = \mathbf{X}\boldsymbol{\beta} + \boldsymbol{\varepsilon} \tag{1}$$

where $\mathbf{Y}$ is an $n \times 1$ vector of response random variables, $\mathbf{X}$ is the design matrix with covariates (fixed explanatory variables, containing any combination of continuous, binary, or categorical variables), $\boldsymbol{\beta}$ is a vector of parameters, and $\boldsymbol{\varepsilon}$ is a vector of zero-mean random variables, which are classically assumed to be uncorrelated, $\text{var}(\boldsymbol{\varepsilon}) = \sigma^2\mathbf{I}$. The spatial linear model is a version of Eq (1) where $\text{var}(\boldsymbol{\varepsilon}) = \mathbf{V}$, and $\mathbf{V}$ is a patterned covariance matrix that is modeled using spatial relationships. Generally, spatial relationships are of two types: spatially-continuous point-referenced data, often called geostatistics, and finite sets of neighbor-based data, often called lattice or areal data [1]. For geostatistical data, we associate random variables in Eq (1) with their spatial locations by denoting the random variable as $Y(\mathbf{s}_i); i = 1, \ldots, n$, and $\varepsilon(\mathbf{s}_i); i = 1, \ldots, n$, where $\mathbf{s}_i$ is a vector of spatial coordinates for the $i$th point, and the $i, j$th element of $\mathbf{V}$ is $\text{cov}(\varepsilon(\mathbf{s}_i), \varepsilon(\mathbf{s}_j))$. Table 1 provides a list of all of the main notation used in this article.

The main goals from a geostatistical linear model are to 1) estimate $\mathbf{V}$, 2) estimate $\boldsymbol{\beta}$, 3) make predictions at unsampled $Y(\mathbf{s}_j)$, where $j = n + 1, \ldots, N$, form a set of spatial locations without observations, and 4) for some region $\mathcal{B}$, make a prediction of the average value $Y(\mathcal{B}) = \int_{\mathcal{B}} Y(\mathbf{s})d\mathbf{s}/|\mathcal{B}|$, where $|\mathcal{B}|$ is the area of $\mathcal{B}$. Estimation and prediction both require $\mathcal{O}(n^2)$ for $\mathbf{V}$ storage and $\mathcal{O}(n^3)$ operations for $\mathbf{V}^{-1}$ [2], which, for massive data sets, is computationally expensive and may be prohibitive. Our overall objective is to use spatial indexing ideas to make all four goals possible for very large spatial data sets. We maintain the moment-based approach of classical geostatistics, which is distribution free, and we work to maintain a coherent model of stationarity and a single set of parameter estimates.

**Table 1. Commonly-used symbols and their meanings in this paper.**

| | |
|---|---|
| $Y(\mathbf{s})$ | response random variable at spatial location $\mathbf{s}$ |
| $\mathbf{s}$ | vector of spatial coordinates |
| $\mathbf{Y}$ | random vector of response variables |
| $\mathbf{y}$ | observed data vector of response variables |
| $\mathbf{X}$ | design matrix of fixed effects |
| $\boldsymbol{\beta}$ | vector of fixed-effect parameters |
| $\boldsymbol{\varepsilon}$ | vector of spatially-autocorrelated random errors |
| $\mathbf{V}$ | covariance matrix of $\boldsymbol{\varepsilon}$ |
| $\boldsymbol{\theta}$ | vector of covariance parameters |
| $\mathbf{c}_0$ | covariance between data and prediction location |
| $\mathcal{L}(\boldsymbol{\theta}; \mathbf{y})$ | likelihood of $\boldsymbol{\theta}$ given data $\mathbf{y}$ |
| $\mathbf{T}_{xx}$ | $\sum_{i=1}^{P} \mathbf{X}_i'\hat{\mathbf{V}}_{i,i}^{-1}\mathbf{X}_i$ |
| $\mathbf{t}_{xy}$ | $\sum_{i=1}^{P} \mathbf{X}_i'\hat{\mathbf{V}}_{i,i}^{-1}\mathbf{y}_i$ |
| $\mathbf{Q}$ | $\left[\mathbf{T}_{xx}^{-1}\mathbf{X}_1\hat{\mathbf{V}}_{1,1}^{-1}|\mathbf{T}_{xx}^{-1}\mathbf{X}_2\hat{\mathbf{V}}_{2,2}^{-1}|\ldots|\mathbf{T}_{xx}^{-1}\mathbf{X}_P\hat{\mathbf{V}}_{P,P}^{-1}\right]$ |
| $\mathbf{W}_{xx}$ | $\sum_{i=1}^{P-1}\sum_{j=i+1}^{P}[\mathbf{X}_i'\mathbf{V}_{i,i}^{-1}\mathbf{V}_{i,j}\mathbf{V}_{j,j}^{-1}\mathbf{X}_j + (\mathbf{X}_i'\mathbf{V}_{i,i}^{-1}\mathbf{V}_{i,j}\mathbf{V}_{j,j}^{-1}\mathbf{X}_j)']$ |
| $\mathbf{N}_j$ | 0-1 matrix to subset $\mathbf{y}$ to the neighborhood of the $j$th location |
| $\hat{\mathbf{C}}$ | an estimator of $\text{var}(\hat{\boldsymbol{\beta}}_{bd})$ |

## Quick review of the spatial linear model

When the outcome of the random variable $Y(\mathbf{s}_i)$ is observed, we denote it $y(\mathbf{s}_i)$, which are contained in the vector $\mathbf{y}$. These observed data are used first to estimate the autocorrelation parameters in $\mathbf{V}$, which we will denote as $\boldsymbol{\theta}$. In general, $\mathbf{V}$ can have $n(n+1)/2$ parameters, but use of distance to describe spatial relationships typically reduces this to just 3 or 4 parameters. An example of how $\mathbf{V}$ depends on $\boldsymbol{\theta}$ is given by the exponential autocorrelation model, where the $i,j$th element of $\mathbf{V}$ is

$$\text{cov}[\boldsymbol{\varepsilon}(\mathbf{s}_i), \boldsymbol{\varepsilon}(\mathbf{s}_j)] = \tau^2 \exp(-d_{i,j}/\rho) + \eta^2 \mathcal{I}(d_{i,j} = 0) \tag{2}$$

where $\boldsymbol{\theta} = (\tau^2, \eta^2, \rho)'$, $d_{i,j}$ is the Euclidean distance between $\mathbf{s}_i$ and $\mathbf{s}_j$, and $\mathcal{I}(\cdot)$ is an indicator function, equal to 1 if its argument is true, otherwise it is 0. The parameter $\eta^2$ is often called the "nugget effect," $\tau^2$ is called the "partial sill," and $\rho$ is called the "range" parameter. In Eq (2), the variances are constant (stationary), which we denote $\sigma^2 = \tau^2 + \eta^2$, when $d_{i,j} = 0$. Many other examples of autocorrelation model are given in [1, 3].

We will use restricted maximum likelihood (REML) [4, 5] to estimate parameters of $\mathbf{V}$. REML is less biased than full maximum likelihood [6]. REML estimates of covariance parameters are obtained by minimizing

$$\mathcal{L}(\boldsymbol{\theta}|\mathbf{y}) = \log|\mathbf{V}_{\boldsymbol{\theta}}| + \mathbf{r}'_{\boldsymbol{\theta}} \mathbf{V}_{\boldsymbol{\theta}}^{-1} \mathbf{r}_{\boldsymbol{\theta}} + \log|\mathbf{X}'\mathbf{V}_{\boldsymbol{\theta}}^{-1}\mathbf{X}| + c \tag{3}$$

for $\boldsymbol{\theta}$, where $\mathbf{V}_{\boldsymbol{\theta}}$ depends on spatial autocorrelation parameters $\boldsymbol{\theta}$, and $\mathbf{r}_{\boldsymbol{\theta}} = \mathbf{y} - \mathbf{X}\hat{\boldsymbol{\beta}}_{\boldsymbol{\theta}}$, $\hat{\boldsymbol{\beta}}_{\boldsymbol{\theta}} = (\mathbf{X}'\mathbf{V}_{\boldsymbol{\theta}}^{-1}\mathbf{X})^{-1}\mathbf{X}'\mathbf{V}_{\boldsymbol{\theta}}^{-1}\mathbf{y}$, and $c$ is a constant that does not depend on $\boldsymbol{\theta}$. It has been shown [7, 8] that Eq (3) form unbiased estimating equations for covariance parameters, so Gaussian data are not strictly necessary. After Eq (3) has been minimized for $\boldsymbol{\theta}$, then these estimates, call them $\hat{\boldsymbol{\theta}}$, are used in the autocorrelation model, e.g. Eq 2, for all of the covariance values to create $\hat{\mathbf{V}}$. This is the first use of data $\mathbf{y}$. The usual frequentist method for geostatistics, with a long tradition [9], "uses the data twice" [10]. Now $\hat{\mathbf{V}}$, along with a second use of the data, are used to estimate regression coefficients or make predictions at unsampled locations. By plugging $\hat{\mathbf{V}}$ into the well-known best-linear-unbiased estimate (BLUE) of $\boldsymbol{\beta}$ for Eq (1), we obtain the empirical best-linear-unbiased estimate (EBLUE), e.g. [11],

$$\hat{\boldsymbol{\beta}} = (\mathbf{X}'\hat{\mathbf{V}}^{-1}\mathbf{X})^{-1}\mathbf{X}'\hat{\mathbf{V}}^{-1}\mathbf{y} \tag{4}$$

The estimated variance of Eq (4) is

$$\hat{\text{var}}(\hat{\boldsymbol{\beta}}) = (\mathbf{X}'\hat{\mathbf{V}}^{-1}\mathbf{X})^{-1} \tag{5}$$

Let a single unobserved location be denoted $\mathbf{s}_0$, with a covariate vector of $\mathbf{x}_0$ (containing the same covariates and length as a row of $\mathbf{X}$). Then empirical best-linear-unbiased prediction (EBLUP) [12] at an unobserved location is

$$\hat{Y}(\mathbf{s}_0) = \mathbf{x}'_0\hat{\boldsymbol{\beta}} + \hat{\mathbf{c}}'_0\hat{\mathbf{V}}^{-1}(\mathbf{y} - \mathbf{X}\hat{\boldsymbol{\beta}}), \tag{6}$$

where $\hat{\mathbf{c}}_0 \equiv \hat{\text{cov}}(\boldsymbol{\varepsilon}, \boldsymbol{\varepsilon}(\mathbf{s}_0))$, using the same autocorrelation model, e.g. Eq (2), and estimated parameters, $\hat{\boldsymbol{\theta}}$, that were used to develop $\hat{\mathbf{V}}$. Note that if we condition on $\hat{\mathbf{V}}$ as fixed, then Eq (6) is a linear combination of $\mathbf{y}$, and can also be written as $\boldsymbol{\eta}'_0\mathbf{y}$ when Eq (4) is substituted for $\hat{\boldsymbol{\beta}}$. The prediction Eq (6) can be seen as the conditional expectation of $Y(\mathbf{s}_0)|\mathbf{y}$ with plug-in values

for $\boldsymbol{\beta}$, $\mathbf{V}$, and $\mathbf{c}$. The estimated variance of EBLUP is,

$$\hat{\text{var}}(\hat{Y}(\mathbf{s}_0)) = \hat{\sigma}_0^2 - \hat{\mathbf{c}}_0'\hat{\mathbf{V}}^{-1}\hat{\mathbf{c}}_0 + (\mathbf{x}_0 - \mathbf{X}'\hat{\mathbf{V}}^{-1}\hat{\mathbf{c}}_0)'(\mathbf{X}'\hat{\mathbf{V}}^{-1}\mathbf{X})^{-1}(\mathbf{x}_0 - \mathbf{X}'\hat{\mathbf{V}}^{-1}\hat{\mathbf{c}}_0) \quad (7)$$

where $\hat{\sigma}_0^2$ is the estimated variance of $Y(\mathbf{s}_0)$ using the same covariance model as $\hat{\mathbf{V}}$. [12]

## Spatial methods for big data

Here, we give a brief overview of the most popular methods currently used for large spatial data sets. There are various ways to classify such methods. For our purposes, there are two broad approaches. One is to adopt a Gaussian Process (GP) model for the data and then approximate the GP. The other is to model locally, essentially creating smaller data sets and using existing models.

There are several good reviews on methods for approximating the GP [13–16]. These methods include low rank ideas such as radial smoothing [17–19], fixed rank kriging [20–23], predictive processes [24, 25], and multiresolution Gaussian processes [26, 27]. Other approaches include covariance tapering [28–30], stochastic partial differential equations [31, 32], and factoring the GP into a series of conditional distributions [33, 34], which was extended to nearest neighbor Gaussian processes [35–38] and other sparse matrix improvements [39–41]. The reduced rank methods are very attractive, and allow models for situations where distances are non-Euclidean (for a review and example, see [42]), as well as fast computation.

Modeling locally involves an attempt to maintain classical geostatistical models by creating subsets of the data, using existing methods on subsets, and then making inference from subsets. For example, [43, 44] created local data sets in a spatial moving window, and then estimated variograms and used ordinary kriging within those windows. This idea allows for nonstationary variances but forces an unnatural asymmetric autocorrelation because the range parameter changes when moving a window. Nor does it estimate $\boldsymbol{\beta}$, but rather there is a different $\boldsymbol{\beta}$ for every point in space. Another early idea was to create a composite likelihood by taking products of subset-likelihoods and optimizing for autocorrelation parameters $\boldsymbol{\theta}$ [45], and then $\hat{\boldsymbol{\theta}}$ can be held fixed when predicting in local windows. However, this does not solve the problem of estimating a single $\boldsymbol{\beta}$.

More recently, two broad approaches have been developed for modeling locally. One is a 'divide and conquer' approach, which is similar to [45]. Here, it is permissible to re-use data in subsets, or not use some data at all [46–48], with an overview provided by [49]. Another approach is a simple partition of the data into groups, where partitions are generally spatially compact [50–53]. This is sensible for estimating covariance parameters and will provide an unbiased estimate for $\hat{\boldsymbol{\beta}}$, however the estimated variance $\hat{\text{var}}(\hat{\boldsymbol{\beta}})$ will not be correct. Continuity corrections for predictions are provided, but predictions may not be efficient near partition boundaries.

A blocked structure for the covariance matrix based on spatially-compact groupings was proposed by [54], who then formulated a hybrid likelihood based on blocks of different sizes. The method that we feature is most similar to [54], but we show that there is no need for a hybrid likelihood, and that our approach is different than composite likelihood. Our spatial indexing approach is very simple and extends easily to random effects, and accommodates virtually any covariance matrix that can be constructed. We also show how to obtain the exact covariance matrix of estimated fixed effects without any need for computational derivatives or numerical approximations.

## Motivating example

One of the attractive features of the method that we propose is that it will work with *any* valid covariance matrix. To motivate our methods, consider a stream network (Fig 1a). This is the Mid-Columbia River basin, located along part of the border between the states of Washington

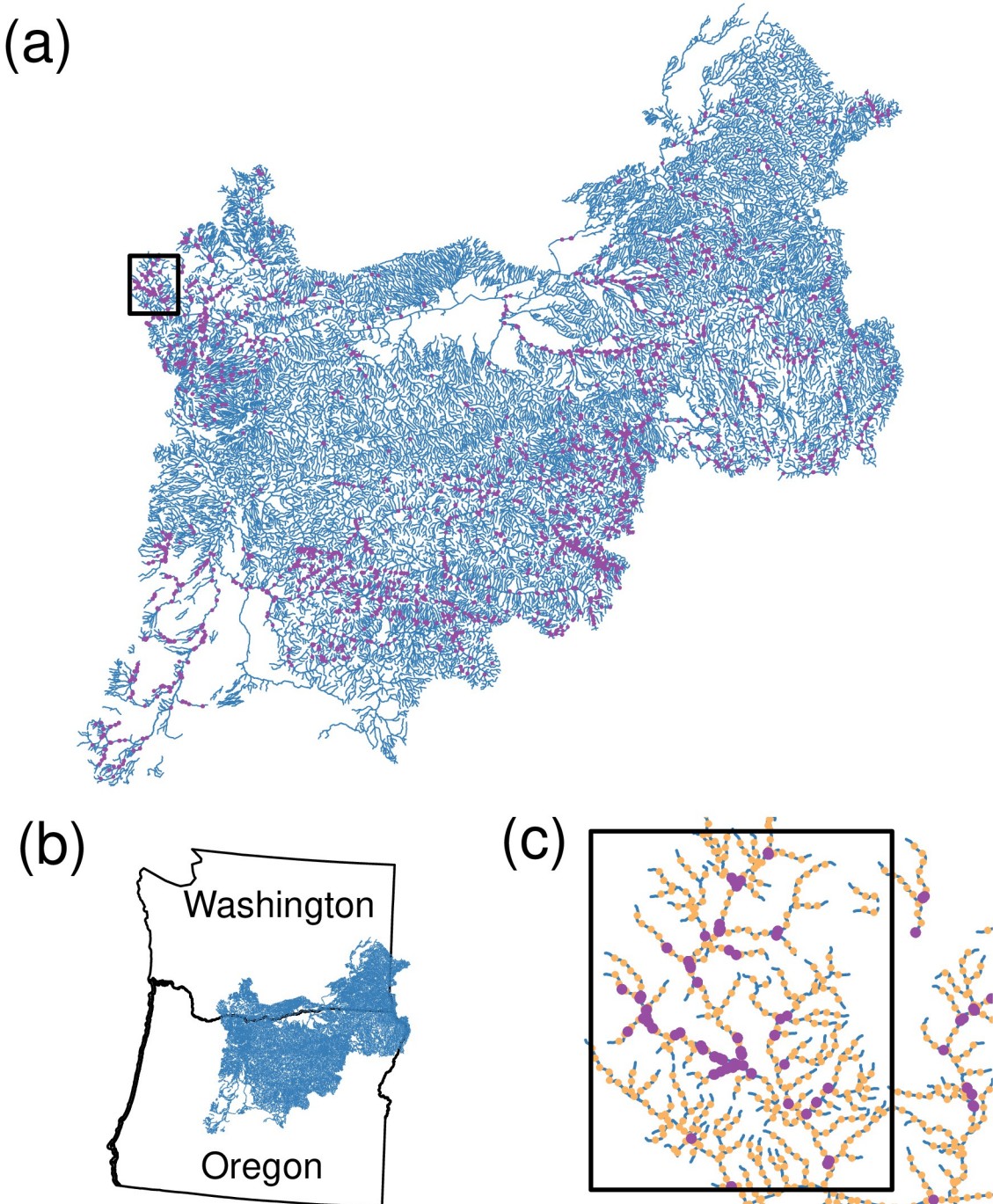

**Fig 1. Study area for the motivating example.** (a) A stream network from the mid-Columbia River basin, where purple points show 9521 sample locations that measured mean water temperature during August. (b) Most of the stream network is located in Washington and Oregon in the United States. (c) A close-up of the black rectangle in (a). The orange points are prediction locations.

and Oregon, USA, with a small part of the network in Idaho as well (Fig 1b). The stream network consists of 28,613 stream segments. Temperature loggers were placed at 9,521 locations on the stream network, indicated by purple dots in Fig 1a. A close-up of the stream network, indicated by the dark rectangle in Fig 1a, is given as Fig 1c, where we also show a systematic placement of prediction locations with orange dots. There are 60,099 prediction locations that will serve as the basis for point predictions. The response variable is an average of daily maximum temperatures in August from 1993 to 2011. Explanatory variables obtained for both observations and prediction sites included elevation at temperature logger site, slope of stream segment at site, percentage of upstream watershed composed of lakes or reservoirs, proportion of upstream watershed composed of glacial ice surfaces, mean annual precipitation in watershed upstream of sensor, the northing coordinate, base-flow index values, upstream drainage area, a canopy value encompassing the sensor, mean August air temperature from a gridded climate model, mean August stream discharge, and occurrence of sensor in tailwater downstream from a large dam (see [55] for more details).

These data were previously analyzed in [55] with geostatistical models specific to stream networks [11, 56]. The models were constructed as spatial moving averages, e.g., [57, 58], also called process convolutions, e.g., [59, 60]. Two basic covariance matrices are constructed, and then summed. In one, random variables were constructed by integrating a kernel over a white noise process strictly upstream of a site, which are termed "tail-up" models. In the other construction, random variables were created by integrating a kernel over a white noise process strictly downstream of a site, which are termed "tail-down" models. Both types of models allow analytical derivation of autocovariance functions, with different properties. For tail-up models, sites remain independent so long as they are not connected by water flow from an upstream site to a downstream site. This is true even if two sites are very close spatially, but each on a different branch just upstream of a junction. Tail-down models are more typical as they allow spatial dependence that is generally a function of distance along the stream, but autocorrelation will still be different for two pairs of sites that are an equal distance apart, when one pair is connected by flow, and the other is not.

When considering big data, such as those in Fig 1, we considered the methods as described in the previous section. The basis-function/reduced rank approaches would be difficult for stream networks because an inspection of Fig 1 reveals that we would need thousands of basis functions in order to cover all headwater stream segments and run the basis functions downstream only. A separate set of basis functions would be needed that ran upstream, and then weighting would be required to split the basis functions at all stream junctions. In fact, all of the GP model approximation methods would require modifying a covariance structure that has already been developed specifically for steam networks. The spatial indexing method that we propose below is much simpler, requiring no modification to the covariance structure, and, as we will demonstrate, proved to be adequate, not only for stream networks, but more generally.

## Objectives

In what is to follow, we will use spatial indexing, leading to covariance matrix partitioning and local predictions. We will use the acronym SPIN, for SPatial INdexing, as the collection of methods for covariance parameter estimation, fixed effects estimation, and point and block prediction. Our objective is to show how each of these inferences can be made computationally faster with SPIN, and still provide unbiased results with valid confidence/prediction intervals.

This article uses several acronyms. Table 2 provides a handy reference to the meaning of all acronyms used here.

**Table 2. Acronyms used in this paper.**

| | |
|---|---|
| ANOVA | analysis of variance |
| BLUE | best linear unbiased estimation |
| CI90 | coverage rates for 90% confidence intervals |
| COMP | spatially compact partitioning |
| COPE | covariance parameter estimation |
| EBLUE | empirical best-linear-unbiased estimation |
| EBLUP | empirical best-linear-unbiased prediction |
| FEFE | fixed-effects parameter estimation |
| GEOSTAT | geostatistical simulation method |
| GP | Gaussian process |
| MIXD | mix of random and spatially compact partitioning |
| NNGP | nearest-neighbor Gaussian processes method |
| PI90 | coverage rates for 90% prediction intervals |
| RAND | random partitioning |
| REML | restricted maximum likelihood |
| RMSE | root mean-squared error |
| RMSPE | root mean-squared-prediction error |
| SPIN | computationally-fast inference methods using spatial indexing |
| SUMSINE | simulation method based on random sine waves |

## Methods

The main advantage of the SPIN method is due to the way the covariance matrix is indexed and partitioned to allow for faster evaluation of the REML equations, Eq (3), whose optimization is iterative, requiring many evaluations involving the inverse of the covariance matrix. This optimization provides estimation of the covariance parameters, which we describe next.

### Estimation of covariance parameters

Consider the covariance matrix to be used in Eqs (4) and (6). First, we index the data to create a covariance matrix with $P$ partitions based on the indexes $\{i; i = 1, \ldots, P\}$,

$$
\mathbf{V} = \begin{pmatrix} \mathbf{V}_{1,1} & \mathbf{V}_{1,2} & \cdots & \mathbf{V}_{1,P} \\ \mathbf{V}_{2,1} & \mathbf{V}_{2,2} & \cdots & \mathbf{V}_{2,P} \\ \vdots & \vdots & \ddots & \vdots \\ \mathbf{V}_{P,1} & \mathbf{V}_{P,2} & \cdots & \mathbf{V}_{P,P} \end{pmatrix}
\tag{8}
$$

In a similar way, imagine a corresponding indexing and partition of the spatial linear model as,

$$
\begin{pmatrix} \mathbf{y}_1 \\ \mathbf{y}_2 \\ \vdots \\ \mathbf{y}_P \end{pmatrix} = \begin{pmatrix} \mathbf{X}_1 \\ \mathbf{X}_2 \\ \vdots \\ \mathbf{X}_P \end{pmatrix} \boldsymbol{\beta} + \begin{pmatrix} \boldsymbol{\varepsilon}_1 \\ \boldsymbol{\varepsilon}_2 \\ \vdots \\ \boldsymbol{\varepsilon}_P \end{pmatrix}
\tag{9}
$$

Now, for the purposes of estimating covariance parameters, we maximize the REML equations

based on a covariance matrix,

$$\mathbf{V}_{part} = \begin{pmatrix} \mathbf{V}_{1,1} & \mathbf{0} & \cdots & \mathbf{0} \\ \mathbf{0} & \mathbf{V}_{2,2} & \cdots & \mathbf{0} \\ \vdots & \vdots & \ddots & \vdots \\ \mathbf{0} & \mathbf{0} & \cdots & \mathbf{V}_{P,P} \end{pmatrix} \tag{10}$$

rather than Eq (8). The computational advantage of using Eq (10) in Eq (3) is that we only need to invert matrices of size $\mathbf{V}_{i,i}$ for all $i$, and, because we have large amounts of data, we assume that $\{\mathbf{V}_{i,i}\}$ are sufficient for estimating covariance parameters. If the size of $\mathbf{V}_{i,i}$ is fixed, then the computational burden grows linearly with $n$. Also, Eq (10) in Eq (3) allows for use of parallel computing because each $\mathbf{V}_{i,i}$ can be inverted independently.

Note that we are not concerned with the variance of $\hat{\boldsymbol{\theta}}$, which is generally true in classical geostatistics. Rather, $\boldsymbol{\theta}$ contains nuisance parameters that require estimation in order to estimate fixed effects and make predictions. Because data are massive, we can afford to lose some efficiency in estimating the covariance parameters. For example, sample sizes $\geq 125$ are generally recommended for estimating the covariance matrix for geostatistical data [61]. REML is for the most part unbiased. If we have thousands of samples, and if we imagine partitioning the spatial locations into data sets (in ways that we describe later), then using Eq (10) in Eq (3) is, essentially, using REML many times to obtain a pooled estimate of $\hat{\boldsymbol{\theta}}$.

Partitioning the covariance matrix is most closely related to the idea of quasi-likelihood [62], composite likelihood [45] and divide and conquer [63]. However, for REML, they are not exactly equivalent. From Eq (3), the term $\log|\mathbf{X}'\mathbf{V}_{\boldsymbol{\theta}}^{-1}\mathbf{X}|$ using composite likelihood, $\prod_{i=1}^{P} \mathcal{L}(\boldsymbol{\theta}|\mathbf{y}_i)$, results in

$$\sum_{i=1}^{P} \log|\mathbf{X}_i'\mathbf{V}_{i,i}^{-1}\mathbf{X}_i|$$

while using $\mathbf{V}_{part}$ results in

$$\log\left|\sum_{i=1}^{P} \mathbf{X}_i'\mathbf{V}_{i,i}^{-1}\mathbf{X}_i\right|$$

An advantage to spatial indexing, when compared to composite likelihood, can be seen when $\mathbf{X}$ contains columns with many zeros, such as may occur for categorical explanatory variables. Then, partitioning $\mathbf{X}$ may result in $\mathbf{X}_i$ that has columns with all zeros, which presents a problem when computing $\log|\mathbf{X}_i'\mathbf{V}_{i,i}^{-1}\mathbf{X}_i|$ for composite likelihood, but not when using $\mathbf{V}_{part}$.

The SPIN indexing can also allow for faster inversion of the covariance matrix when estimating fixed effects, but that requires some new results to obtain the proper standard errors of the estimated fixed effects, which we describe next.

## Estimation of $\boldsymbol{\beta}$

The generalized least squares estimate for $\boldsymbol{\beta}$ was given in Eq (4). Although the inverse $\mathbf{V}^{-1}$ only occurs once (as compared to repeatedly when optimizing the REML equations), it will still be computationally prohibitive if a data set has thousands of samples. Note that under the partitioned model, Eq (9) with covariance matrix Eqs (10), (4), is,

$$\hat{\boldsymbol{\beta}}_{bd} = \mathbf{T}_{xx}^{-1}\mathbf{t}_{xy} \tag{11}$$

where $\mathbf{T}_{xx} = \sum_{i=1}^{P} \mathbf{X}'_i \hat{\mathbf{V}}_{i,i}^{-1} \mathbf{X}_i$ and $\mathbf{t}_{xy} = \sum_{i=1}^{P} \mathbf{X}'_i \hat{\mathbf{V}}_{i,i}^{-1} \mathbf{y}_i$. This is a "pooled estimator" of $\boldsymbol{\beta}$ across the partitions. This should be a good estimator of $\boldsymbol{\beta}$ at a much reduced computational cost. It will also be convenient to show that Eq (11) is linear in $\mathbf{y}$, by noting that

$$\hat{\boldsymbol{\beta}}_{bd} = \left[ \mathbf{T}_{xx}^{-1} \mathbf{X}_1 \hat{\mathbf{V}}_{1,1}^{-1} | \mathbf{T}_{xx}^{-1} \mathbf{X}_2 \hat{\mathbf{V}}_{2,2}^{-1} | \ldots | \mathbf{T}_{xx}^{-1} \mathbf{X}_P \hat{\mathbf{V}}_{P,P}^{-1} \right] \begin{bmatrix} \mathbf{y}_1 \\ \mathbf{y}_2 \\ \vdots \\ \mathbf{y}_P \end{bmatrix} = \mathbf{Q}\mathbf{y} \tag{12}$$

To estimate the variance of $\hat{\boldsymbol{\beta}}_{bd}$ we cannot ignore the correlation between the partitions, so we consider the full covariance matrix Eq (8). If we compute the covariance matrix for Eq (11) under the full covariance matrix Eq (8), we obtain

$$\hat{\mathrm{var}}(\hat{\boldsymbol{\beta}}_{bd}) = \mathbf{T}_{xx}^{-1} + \mathbf{T}_{xx}^{-1} \mathbf{W}_{xx} \mathbf{T}_{xx}^{-1} \tag{13}$$

where $\mathbf{W}_{xx} = \sum_{i=1}^{P-1} \sum_{j=i+1}^{P} [\mathbf{X}'_i \mathbf{V}_{i,i}^{-1} \mathbf{V}_{i,j} \mathbf{V}_{j,j}^{-1} \mathbf{X}_j + (\mathbf{X}'_i \mathbf{V}_{i,i}^{-1} \mathbf{V}_{i,j} \mathbf{V}_{j,j}^{-1} \mathbf{X}_j)']$. Note that while we set parts of $\mathbf{V} = \mathbf{0}$ Eq (10) in order to estimate $\boldsymbol{\theta}$ and $\boldsymbol{\beta}$, we computed the variance of $\hat{\boldsymbol{\beta}}$ using the full $\mathbf{V}$ in Eq (8). Using a plug-in estimator, whereby $\boldsymbol{\theta}$ is replaced by $\hat{\boldsymbol{\theta}}$, no further inverses of any $\mathbf{V}_{i,j}$ are required if all $\mathbf{V}_{i,i}^{-1}$ are stored as part of the REML optimization. There is only a single additional inverse required, which is $R \times R$, where $R$ is the rank of the design matrix $\mathbf{X}$, and is already computed for $\mathbf{T}_{xx}^{-1}$ in Eq (11). Also note that if we simply substituted Eq (10) into Eq (5), then we obtain only $\mathbf{T}_{xx}^{-1}$ as the variance of $\hat{\boldsymbol{\beta}}_{bd}$. In Eq (13), $\mathbf{T}_{xx}^{-1} \mathbf{W}_{xx} \mathbf{T}_{xx}^{-1}$ is the adjustment that is required for correlation among the partitions for a pooled estimate of $\hat{\boldsymbol{\beta}}_{bd}$. Partitioning of the spatial linear model allows computation from Eq (11), but then going back to the full model for developing Eq (13), which is a new result. This can be contrasted to the approaches for variance estimation of fixed effects using pseudo likelihood, composite likelihood, and divide and conquer found in the earlier literature review.

Eq (13) is quite fast and grows linearly for computing the number of inverse matrices $\mathbf{V}_{i,i}^{-1}$ (that is, if observed sample size is $2n$, then there are twice as many inverses as a sample of size $n$, if we hold partition size fixed). Also note that all inverses may already be computed as part of REML estimation of $\boldsymbol{\theta}$. However, Eq (13) is quadratic in pure matrix computations due to the double sum in $\mathbf{W}_{xx}$. These can be made parallel, but may take too long for more than about 100,000 samples. One alternative is to use the empirical variation in $\hat{\boldsymbol{\beta}}_i = (\mathbf{X}'_i \hat{\mathbf{V}}_{i,i}^{-1} \mathbf{X}_i)^{-1} \mathbf{X}'_i \hat{\mathbf{V}}_{i,i}^{-1} \mathbf{y}_i$, where the $i$th matrix calculations are already needed for Eq (11) and $\hat{\boldsymbol{\beta}}_i$ can be simply computed and stored. Then, let

$$\hat{\mathrm{var}}_{alt1}(\hat{\boldsymbol{\beta}}_{bd}) = \frac{1}{P(P-1)} \sum_{i=1}^{P} (\hat{\boldsymbol{\beta}}_i - \hat{\boldsymbol{\beta}}_{bd})(\hat{\boldsymbol{\beta}}_i - \hat{\boldsymbol{\beta}}_{bd})' \tag{14}$$

which has been used before for partitioned data, e.g. [64]. A second alternative is to pool the estimated variances of each $\hat{\boldsymbol{\beta}}_i$, which are $\hat{\mathrm{var}}(\hat{\boldsymbol{\beta}}_i) = (\mathbf{X}'_i \hat{\mathbf{V}}_{i,i}^{-1} \mathbf{X}_i)^{-1}$, to obtain

$$\hat{\mathrm{var}}_{alt2}(\hat{\boldsymbol{\beta}}_{bd}) = \frac{1}{P^2} \sum_{i=1}^{P} \hat{\mathrm{var}}(\hat{\boldsymbol{\beta}}_i) \tag{15}$$

where the first $P$ in the denominator is for averaging individual $\hat{\mathrm{var}}(\hat{\boldsymbol{\beta}}_i)$, and the second $P$ is

the reduction in variance due to averaging $\hat{\boldsymbol{\beta}}_i$. Eqs (13)–(15) are tested and compared below using simulations.

## Point prediction

The predictor for $Y(\mathbf{s}_0)$ was given in Eq (6). As for estimation, the inverse $\mathbf{V}^{-1}$ only occurs once (as compared to repeatedly when optimizing to obtain the REML estimates). If the data set has tens of thousands of samples, it will still be computationally prohibitive. Note that under the partitioned model, Eq (9), that assumes zero correlation among partitions, Eq (10), from Eq (6) the predictor is,

$$\hat{Y}(\mathbf{s}_0) = \mathbf{x}_0'\hat{\boldsymbol{\beta}}_{bd} + t_{cy} - \mathbf{t}_{xc}'\hat{\boldsymbol{\beta}}_{bd} \tag{16}$$

where $\hat{\boldsymbol{\beta}}_{bd}$ is obtained from Eq (11), $t_{cy} = \sum_{i=1}^{P} \hat{\mathbf{c}}_i'\mathbf{V}_{i,i}^{-1}\mathbf{y}_i$, $\mathbf{t}_{xc} = \sum_{i=1}^{P} \mathbf{X}_i'\mathbf{V}_{i,i}^{-1}\hat{\mathbf{c}}_i$, and $\hat{\mathbf{c}}_i = \hat{\mathrm{cov}}(Y(\mathbf{s}_0), \mathbf{y}_i)$, using the same autocorrelation model and parameters as for $\hat{\mathbf{V}}$. Even though the predictor is developed under the block diagonal matrix Eq (10), the true prediction variance can be computed under Eq (8), as we did for estimation. However, the performance of these predictors turned out to be quite poor.

   We recommend point predictions based on local data instead, which is an old idea, e.g. [43], and has already been implemented in software for some time, e.g. [10]. The local data may be in the form of a spatial limitation, such as a radius around the prediction point, or by using a fixed number of nearest neighbors. For example, the R [65] package `nabor` [66] finds nearest neighbors among hundreds of thousands of samples very quickly. Our method will be to use a single set of global covariance parameters as estimated under the covariance matrix partition Eq (10), and then predict with a fixed number of nearest neighbors. We will investigate the effect due to the number of nearest neighbors through simulation.

   A purely local predictor lacks model coherency, as discussed in the literature review section. We use a single $\hat{\boldsymbol{\theta}}$ for covariance, but there is still the issue of $\hat{\boldsymbol{\beta}}$. As seen in Eq (6), estimation of $\boldsymbol{\beta}$ is implicit in the prediction equations. If $\mathbf{y}_j \subset \mathbf{y}$ are data in the neighborhood of prediction location $\mathbf{s}_j$, then using Eq (6) with local $\mathbf{y}_j$ is implicitly adopting a varying coefficient model for $\hat{\boldsymbol{\beta}}$, making it also local, so call it $\hat{\boldsymbol{\beta}}_j$, and it will vary for each prediction location $\mathbf{s}_j$. A further issue arises if there are categorical covariates. It is possible that a level of the covariate is not present in the local neighborhood, so some care is needed to collapse any columns in the design matrix that are all zeros. These are some of the issues that call to question the "coherency" of a model when predicting locally.

   Instead, as for estimating the covariance parameters, we will assume that the goal is to have a single global estimate of $\boldsymbol{\beta}$. Then we take as our predictor for the $j$th prediction location,

$$\hat{Y}_\ell(\mathbf{s}_j) = \mathbf{x}_j'\hat{\boldsymbol{\beta}}_{bd} + \hat{\mathbf{c}}_j'\hat{\mathbf{V}}_j^{-1}(\mathbf{y}_j - \mathbf{X}_j\hat{\boldsymbol{\beta}}_{bd}) \tag{17}$$

where $\mathbf{X}_j$ and $\hat{\mathbf{V}}_j$ are the design and covariance matrices, respectively, for the same neighborhood as $\mathbf{y}_j$, $\mathbf{x}_j$ is a vector of covariates at prediction location $j$, $\hat{\mathbf{c}}_j = \mathrm{cov}(Y(\mathbf{s}_j), \mathbf{y}_j)$ (using the same autocorrelation model and parameters as for $\hat{\mathbf{V}}_j$), and $\hat{\boldsymbol{\beta}}_{bd}$ was given in Eq (11). It will be convenient for block kriging to note that if we condition on $\hat{\mathbf{V}}_j$ being fixed, then Eq (17) can be written as a linear combination of $\mathbf{y}$, call it $\boldsymbol{\lambda}_j'\mathbf{y}$, similar to $\boldsymbol{\eta}_0'\mathbf{y}$ as mentioned after Eq (6). Suppose there are $m$ neighbors around $\mathbf{s}_j$, so $\mathbf{y}_j$ is $m \times 1$. Let $\mathbf{y}_j = \mathbf{N}_j\mathbf{y}$, where $\mathbf{N}_j$ is a $m \times n$ matrix of

zeros and ones that subset the $n \times 1$ vector of all data to only those in the neighborhood. Then

$$\hat{Y}_\ell(\mathbf{s}_j) = \mathbf{x}_j'\mathbf{Q}\mathbf{y} + \hat{\mathbf{c}}_j'\hat{\mathbf{V}}_j^{-1}\mathbf{N}_j\mathbf{y} - \hat{\mathbf{c}}_j'\hat{\mathbf{V}}_j^{-1}\mathbf{X}_j\mathbf{Q}\mathbf{y} = \boldsymbol{\lambda}_j'\mathbf{y} \tag{18}$$

where $\mathbf{Q}$ was defined in Eq (12).

Let $\hat{\mathbf{C}}$ be an estimator of $\text{var}(\hat{\boldsymbol{\beta}}_{bd})$ in Eqs (13), (14), or 15), then the prediction variance of Eq (17) is $\text{var}(Y(\mathbf{s}_j) - \hat{Y}_\ell(\mathbf{s}_j))$ when using the local neighborhood set of data, which is

$$
\begin{aligned}
\hat{\text{var}}(\hat{Y}_\ell(\mathbf{s}_j)) = & \ \text{E}_{\hat{\boldsymbol{\beta}}_{bd}}\left[\text{var}_{\{\mathbf{y}_j, Y(\mathbf{s}_j)\}}\left(Y(\mathbf{s}_j) - \mathbf{x}_j'\hat{\boldsymbol{\beta}}_{bd} - \hat{\mathbf{c}}_j'\hat{\mathbf{V}}_j^{-1}(\mathbf{y}_j - \mathbf{X}_j\hat{\boldsymbol{\beta}}_{bd})|\hat{\boldsymbol{\beta}}_{bd}\right)\right] + \\
& \ \text{var}_{\hat{\boldsymbol{\beta}}_{bd}}\left[\text{E}_{\{\mathbf{y}_j, Y(\mathbf{s}_j)\}}\left(Y(\mathbf{s}_j) - \mathbf{x}_j'\hat{\boldsymbol{\beta}}_{bd} + \hat{\mathbf{c}}_j'\hat{\mathbf{V}}_j^{-1}(\mathbf{y}_j - \mathbf{X}_j\hat{\boldsymbol{\beta}}_{bd})|\hat{\boldsymbol{\beta}}_{bd}\right)\right] \\
= & \ \hat{\sigma}^2 - \hat{\mathbf{c}}_j'\hat{\mathbf{V}}_j^{-1}\hat{\mathbf{c}}_j + (\mathbf{x}_j - \mathbf{X}_j'\hat{\mathbf{V}}_j^{-1}\hat{\mathbf{c}}_j)'\hat{\mathbf{C}}(\mathbf{x}_j - \mathbf{X}_j'\hat{\mathbf{V}}_j^{-1}\hat{\mathbf{c}}_j)
\end{aligned}
\tag{19}
$$

where $\hat{\sigma}^2$ is the estimated value of $\text{var}(Y(\mathbf{s}_j))$ using $\hat{\boldsymbol{\theta}}$ and the same autocorrelation model that was used for $\hat{\mathbf{V}}$. Eq (19) can be compared to Eq (7).

## Block prediction

None of the literature reviewed earlier considered block prediction, yet it is an important goal in many applications. In fact, the origins of kriging were founded on estimating total gold reserves in the pursuit of mining [9]. The goal of block prediction is to predict the average value over a region, rather than at a point. If that region is a compact set of points denoted as $\mathcal{B}$, then the random quantity is

$$Y(\mathcal{B}) = \frac{1}{|\mathcal{B}|}\int_\mathcal{B} Y(\mathbf{s})d\mathbf{s} \tag{20}$$

where $|\mathcal{B}| = \int_\mathcal{B}1 \ d\mathbf{s}$ is the area of $\mathcal{B}$. In practice, we approximate the integral by a dense set of points on a regular grid within $\mathcal{B}$. Let us call that dense set of points $\mathcal{D} = \{\mathbf{s}_j; j = n + 1, \ldots, N\}$, where recall that $\{\mathbf{s}_j; j = 1, \ldots, n\}$ are the observed data. Then the grid-based approximation to Eq (20) is $Y_\mathcal{D} = (1/N)\sum_{j\in\mathcal{D}}Y(\mathbf{s}_j)$ with generic predictor

$$\hat{Y}_\mathcal{D} = \frac{1}{N}\sum_{j\in\mathcal{D}}\hat{Y}(\mathbf{s}_j)$$

We are in the same situation as for prediction of single sites, where we are unable to invert the covariance matrix of all $n$ observed locations for predicting $\{\hat{Y}(\mathbf{s}_j); j = n + 1, n + 2, \ldots, N\}$. Instead, let us use the local predictions as developed in the previous section, which we will average to compute the block prediction. Let the point predictions be a set of random variables denoted as $\{\hat{Y}_\ell(\mathbf{s}_j), j = n + 1, n + 2, \ldots, N\}$. Denote $\mathbf{y}_o$ a vector of random variables for observed locations, and $\mathbf{y}_u$ a vector of unobserved random variables on the prediction grid $\mathcal{D}$ to be used as an approximation to the block. Recall that we can write Eq (18) as $\hat{Y}_\ell(\mathbf{s}_j) = \boldsymbol{\lambda}_j'\mathbf{y}_o$. We can put all $\boldsymbol{\lambda}_j$ into a large matrix,

$$\mathbf{W} = \begin{pmatrix} \boldsymbol{\lambda}_1' \\ \boldsymbol{\lambda}_2' \\ \vdots \\ \boldsymbol{\lambda}_N' \end{pmatrix}_{(N-n)\times n}$$

The average of all predictions, then, is

$$\hat{Y}_{\mathcal{D}} = \mathbf{a}'\mathbf{W}\mathbf{y}_o \tag{21}$$

where $\mathbf{a} = (1/N, 1/N, \ldots, 1/N)'$. Let $\mathbf{a}'_* = \mathbf{a}'\mathbf{W}$, and so the block prediction $\mathbf{a}'_*\mathbf{y}_o$ is also linear in $\mathbf{y}_o$.

Let the covariance matrix for the vector $(\mathbf{y}'_o, \mathbf{y}'_u)'$ be

$$\mathbf{V} = \begin{pmatrix} \mathbf{V}_{o,o} & \mathbf{V}_{o,u} \\ \mathbf{V}_{u,o} & \mathbf{V}_{u,u} \end{pmatrix}$$

where $\mathbf{V}_{o,o} = \mathbf{V}$ in Eq (8). Then, assuming unbiasedness, that is,
$\mathrm{E}(\mathbf{a}'_*\mathbf{y}_o) = \mathrm{E}(\mathbf{a}'\mathbf{y}_u) \Rightarrow \mathbf{a}_*\mathbf{X}_o\boldsymbol{\beta} = \mathbf{a}\mathbf{X}_u\boldsymbol{\beta}$, where $\mathbf{X}_o$ and $\mathbf{X}_u$ are the design matrices for the observed and unobserved variables, respectively, then the block prediction variance is

$$\mathrm{E}(\mathbf{a}'_*\mathbf{y}_o - \mathbf{a}'\mathbf{y}_u)^2 = \mathbf{a}'_*\mathbf{V}_{o,o}\mathbf{a}_* - 2\mathbf{a}'_*\mathbf{V}_{o,u}\mathbf{a} + \mathbf{a}'\mathbf{V}_{u,u}\mathbf{a} \tag{22}$$

Although the various parts of $\mathbf{V}$ can be very large, the necessary vectors can be created on-the-fly to avoid creating and storing the whole matrix. For example, take the third term in Eq (22). To make the $k$th element of vector $\mathbf{V}_{u,u}\mathbf{a}$, we can create the $k$th row of $\mathbf{V}_{u,u}$, and then take the inner product with $\mathbf{a}$. This means that only the vector $\mathbf{V}_{u,u}\mathbf{a}$ must be stored. We then simply take this vector as an inner product with $\mathbf{a}$ to obtain $\mathbf{a}'\mathbf{V}_{u,u}\mathbf{a}$. Also note that computing Eq (21) grows linearly with observed sample size $n$ due to fixing the number of neighbors used for prediction, but Eq (22) grows quadratically, in both $n$ and $N$, simply due to the matrix dimensions in $\mathbf{V}_{o,o}$ and $\mathbf{V}_{u,u}$. We can control the growth of $N$ by choosing the density of the grid approximation, but it may require subsampling of $\mathbf{y}_o$ if the number of observed data is too large. We often have very precise estimates of block averages, so this may not be too onerous if we have hundreds of thousands of observations.

## The SPIN method

As we have shown, SPIN is a collection of methods for covariance parameter estimation, fixed effects estimation, and point and block prediction, based on spatial indexing. SPIN, as described above, estimates covariance parameters using REML, given by Eq (3), with a valid autocovariance model [e.g., Eq (2) used in a partitioned covariance matrix, given by Eq (10)]. Using these estimated covariance parameters, we estimate $\boldsymbol{\beta}$ using Eq (11), with estimated covariance matrix, Eq (13), unless explicitly stating the use of Eqs (14) or (15). For point prediction, we use Eq (17) with estimated variance Eq (19), unless explicitly stating the purely local version for $\hat{\boldsymbol{\beta}}$ given by Eq (6) with estimated variance Eq (7). For block prediction, we use Eq (21) with Eq (22).

## Simulations

To test the validity of SPIN, we simulated $n$ spatial locations randomly within the $[0, 1] \times [0, 1]$ unit square to be used as observations, and we created a uniformly-spaced $(N - n) = 40 \times 40$ prediction grid within the unit square.

We simulated data with two methods. The first simulation method created data sets that were not actually very large, using exact geostatistical methods that require the Cholesky decomposition of the covariance matrix. For these simulations, we used the spherical autocovariance model to construct $\mathbf{V}$,

$$\mathrm{cov}[\varepsilon(\mathbf{s}_i), \varepsilon(\mathbf{s}_j)] = \tau^2\left(1 - \frac{3d_{i,j}}{2\rho} + \frac{d_{i,j}^3}{2\rho^3}\right)\mathcal{I}(d_{i,j} < \rho) + \eta^2\mathcal{I}(d_{i,j} = 0) \tag{23}$$

where terms are defined as in Eq (2). To simulate normally-distributed data from N($\mathbf{0}$, $\mathbf{V}$), let $\mathbf{L}$ be the lower triangular matrix such that $\mathbf{V} = \mathbf{LL}'$. If vector $\mathbf{z}$ is simulated as independent standard normal variables, then $\boldsymbol{\varepsilon} = \mathbf{Lz}$ is a simulation from N($\mathbf{0}$, $\mathbf{V}$). Unfortunately, computing $\mathbf{L}$ is an $\mathcal{O}(n^3)$ algorithm, on the same order as inverting $\mathbf{V}$, which limits the size of data for simulation. Fig 2a and 2b shows two realizations from N($\mathbf{0}$, $\mathbf{V}$), where the sample size was $n = 2000$ and the autocovariance model, Eq (23), had a $\tau^2 = 10$, $\rho = 0.5$, and $\eta^2 = 0.1$. Each simulation took about 3 seconds. Note that when including evaluation of predictions, simulations are

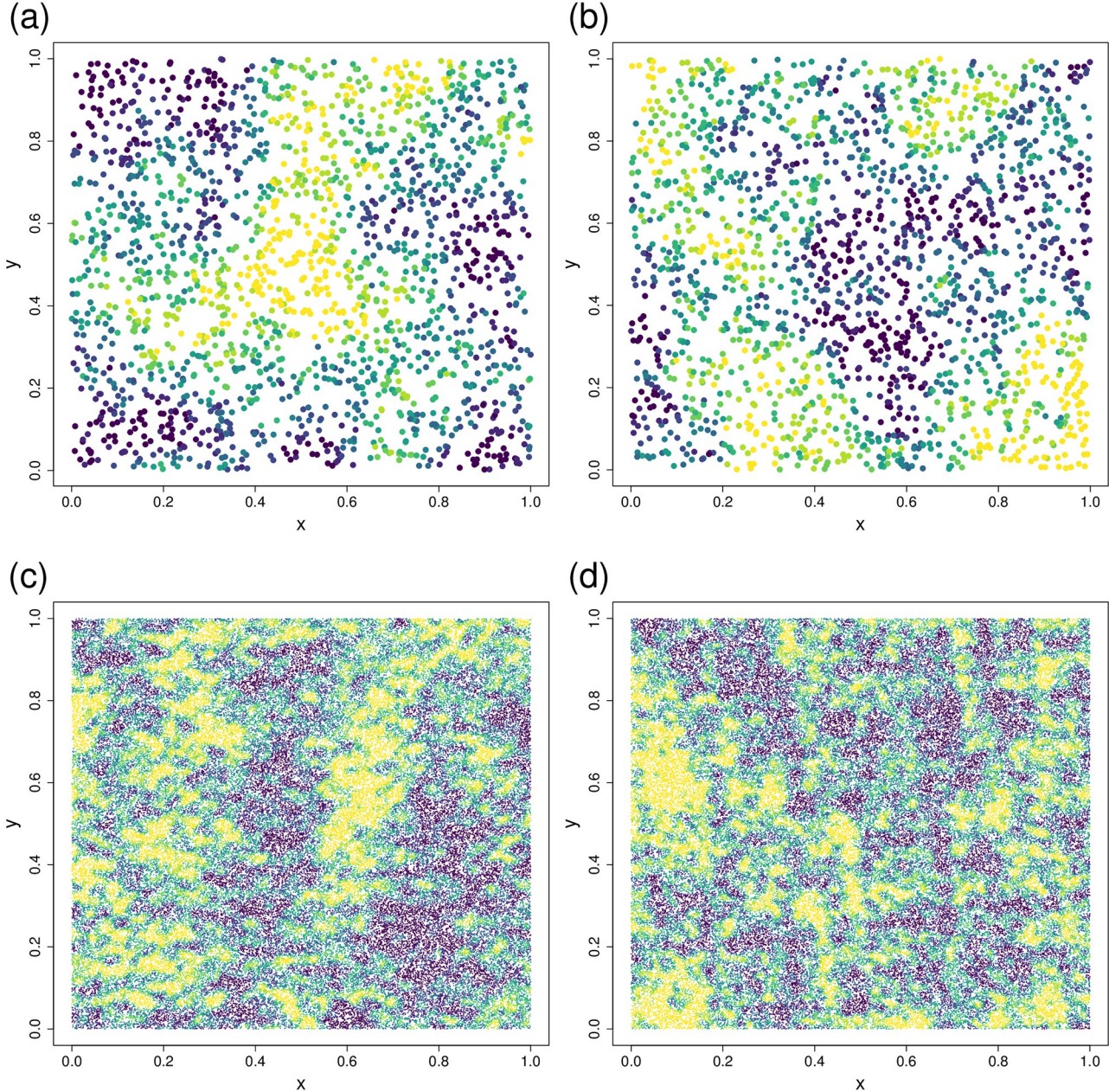

**Fig 2. Examples of simulated surfaces used to test methods.** (a) and (b) are two different realizations of 2000 values from the GEOSTAT method with a range of 2. (c) and (d) are two realizations of 100,000 values from the SUMSINE method. Bluer values are lower, and yellower areas are higher.

required at all $N$ spatial locations. We call this the GEOSTAT simulation method. For all simulations, we fixed $\tau^2 = 10$ and $\eta^2 = 0.1$, but allowed $\rho$ to vary randomly from a uniform distribution between 0 and 2.

We created another method for simulating spatially patterned data for up to several million records. Let $\mathbf{S} = [\mathbf{s}_1, \mathbf{s}_2]$ be the 2-column matrix of the spatial coordinates of data, where $\mathbf{s}_1$ is the first coordinate, and $\mathbf{s}_2$ is the second coordinate. Let

$$\mathbf{S}^* = [\mathbf{s}_1^*, \mathbf{s}_2^*] = \mathbf{S} \begin{bmatrix} \cos(U_{1,i}\pi) & -\sin(U_{1,i}\pi) \\ \sin(U_{1,i}\pi) & \cos(U_{1,i}\pi) \end{bmatrix}$$

be a random rotation of the coordinate system by radian $U_{1,i}\pi$, where $U_{1,i}$ is a uniform random variable. Then let

$$\boldsymbol{\varepsilon}_i = U_{2,i}\left(1 - \frac{i-1}{100}\right)\left[\sin(iU_{3,i}2\pi[\mathbf{s}_1^* + U_{4,i}\pi]) + \sin(iU_{5,i}2\pi[\mathbf{s}_2^* + U_{6,i}\pi])\right] \tag{24}$$

which is a 2-dimensional sine wave surface with a random amplitude (due to uniform random variable $U_{2,i}$), random frequencies on each coordinate (due to uniform random variables $U_{3,i}$ and $U_{5,i}$), and random shifts on each coordinate (due to uniform random variables $U_{4,i}$ and $U_{6,i}$). Then the response variable is created by taking $\boldsymbol{\varepsilon} = \sum_{i=1}^{100}\boldsymbol{\varepsilon}_i$, where expected amplitudes decrease linearly, and expected frequencies increase, with each $i$. Further, the $\boldsymbol{\varepsilon}$ were standardized to zero mean and a variance of 10 for each simulation, and we added a small independent component with variance of 0.1 to each location, similar to the nugget effect $\eta^2$ for the GEOSTAT method. Fig 2c and 2d shows two realizations from the sum of random sine-wave surfaces, where the sample size was 100,000. Each simulation took about 2 seconds. We call this the SUMSINE simulation method.

Thus, random errors, $\boldsymbol{\varepsilon}$, for the simulations were based on GEOSTAT or SUMSINE methods. In either case, we created two fixed effects. A covariate, $x_1(\mathbf{s}_i)$, was generated from standard independent normal-distributions at the $\mathbf{s}_i$ locations. A second spatially-patterned covariate, $x_2(\mathbf{s}_i)$, was created, using the same model, but a different realization, as the random error simulation for $\boldsymbol{\varepsilon}$. Then the response variable was created as,

$$Y(\mathbf{s}_i) = \beta_0 + \beta_1 x_1(\mathbf{s}_i) + \beta_2 x_2(\mathbf{s}_i) + \varepsilon(\mathbf{s}_i) \tag{25}$$

for $i = 1, 2, \ldots$, for a specified sample size $n$, or $N$ (if wanting simulations at prediction sites), and $\beta_0 = \beta_1 = \beta_2 = 1$.

## Evaluation of simulation results

For one summary of performance of fixed effects estimation, we consider the simulation-based estimator of root-mean-squared error,

$$\text{RMSE} = \sqrt{\frac{1}{K}\sum_{k=1}^{K}(\hat{\beta}_{p,k} - \beta_p)^2}$$

for the $k$th simulation among $K$, where $\hat{\beta}_{p,k}$ is the $k$th simulation estimate for the $p$th $\boldsymbol{\beta}$ parameter, and $\beta_p$ is the true parameter used in simulations. We only consider $\beta_1$ and $\beta_2$ in Eq (25). The next simulation-based estimator we consider is 90% confidence interval coverage,

$$\text{CI90} = \frac{1}{K}\sum_{k=1}^{K}\mathcal{I}\left(\hat{\beta}_{p,k} - 1.645\sqrt{\hat{\text{var}}(\hat{\beta}_{p,k})} < \beta_p < \hat{\beta}_{p,k} + 1.645\sqrt{\hat{\text{var}}(\hat{\beta}_{p,k})}\right)$$

To evaluate point prediction we also consider the simulation-based estimator of root-mean-squared prediction error,

$$\text{RMSPE} = \sqrt{\frac{1}{K \times 1600} \sum_{k=1}^{K} \sum_{j=1}^{1600} \left( \hat{Y}_k(\mathbf{s}_j) - y_k(\mathbf{s}_j) \right)^2}$$

where $\hat{Y}_k(\mathbf{s}_j)$ is the predicted value at the $j$th location for the $k$th simulation and $y_k(\mathbf{s}_j)$ is the realized value at the $j$th location for the $k$th simulation. The final summary that we consider is 90% prediction interval coverage,

$$\text{PI90} = \frac{1}{K \times 1600} \sum_{k=1}^{K} \sum_{j=1}^{1600} \mathcal{I} \left( \hat{Y}_k(\mathbf{s}_j) - 1.645 \sqrt{\hat{\text{var}}(\hat{Y}_k(\mathbf{s}_j))} < y_k(\mathbf{s}_j) < \hat{Y}_k(\mathbf{s}_j) + 1.645 \sqrt{\hat{\text{var}}(\hat{Y}_k(\mathbf{s}_j))} \right)$$

where $\hat{\text{var}}(\hat{Y}_k(\mathbf{s}_j))$ is an estimator of the prediction variance.

## Effect of partition method

We wanted to test SPIN over a wide range of data. Hence, we simulated 1000 data sets where simulation method was chosen randomly, with equal probability, between GEOSTAT and SUMSINE methods. If GEOSTAT was chosen, a random sample size between 1000 and 2000 was generated. If SUMSINE was chosen, a random sample size between 2000 and 10,000 was generated. Thus, throughout the study, the simulations occurred over a wide range of parameters, with two different simulation methods and randomly varying autocorrelation. In all cases, the error models fitted to the data were misspecified, because we fitted an exponential autocorrelation model to the true models, GEOSTAT and SUMSINE, that generated them. This should provide a good test of the robustness of the SPIN method and provide fairly general conclusions on the effect of partition method.

After simulating the data, we considered 3 indexing methods. One was completely random, the second was spatially compact, and the third was a mixed strategy, starting with compact, and then 10% were randomly reassigned. To create compact data partitions, we used k-means clustering [67] on the spatial coordinates. K-means has the property of minimizing within group variances and maximizing among group variances. When applied to spatial coordinates, k-means creates spatially compact partitions. An example of each partition method is given in Fig 3. We created partition sizes that ranged randomly from a target of 25 to 225 locations per group (k-means has some variation in group size). It is possible to create one partition for covariance estimation, and another partition for estimating fixed effects. Therefore we considered all nine combinations of the three partition methods for each estimation method.

Table 3 shows performance summaries for the three partition methods, for both fixed effect estimation and point prediction, over wide-ranging simulations when using SPIN with 50 nearest-neighbors for predictions. It is clear that, whether for fixed effect estimation, or prediction, the use of compact partitions was the best option. The worst option was random partitioning. The mixed approach was often close to compact partitioning in performance.

## Effect of partition size

Next, we investigated the effect of partition size. We only used compact partitions, because they were best, and we used partition sizes of 25, 50, 100, and 200 for both covariance parameter estimation and fixed effect estimation, and again used 50 nearest-neighbors for predictions. We simulated data in the same way as above, and used the same performance summaries. Here, we also included the average time, in seconds, for each estimator. The results are shown

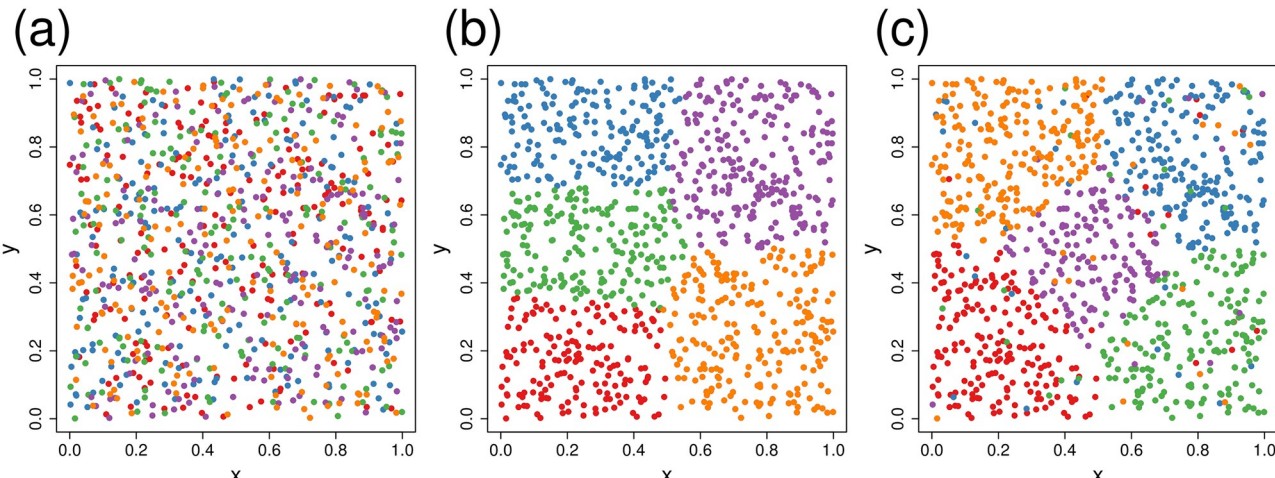

**Fig 3. Illustration of three methods for partitioning data.** Sample size was 1000, and the data were partitioned into 5 groups of 200 each. (a) Random assignment to group. (b) K-means clustering on x- and y-coordinates. (c) K-means on x- and y-coordinates, with 10% randomly re-assigned from each group. Each color represents a different grouping.

in Table 4. In general, larger partition sizes had better RMSE for estimating covariance parameters, but the gains were very small after size 50. For fixed effects estimation, partition size of 50 was often better than 100, and approximately equal to size 200. For prediction, RMSPE was lower as partition size increased. In terms of computing speed, covariance parameter estimation was slower as partition size increased, but fixed effect estimation was faster as partition size increased (because of fewer loops in Eq (13). Partition sizes of 25 often had poor coverage in terms of both CI90 and PI90, but coverage was good for other partition sizes. Based on Tables 3 and 4, one good overall strategy is to use compact partitions of block size 50 for covariance parameter estimation, and block size 200 for fixed effect estimation, for both efficiency and speed. Note that when partition size is different for fixed effect estimation from covariance parameter estimation, new inverses of diagonal blocks in Eq (10) are needed. If

**Table 3. Effect of partition method.**

| COPE | FEFE | RMSE$_1$ | RMSE$_2$ | RMSPE | CI90$_1$ | CI90$_2$ | PI90 |
|------|------|-------|-------|-------|-------|-------|------|
| RAND | RAND | 0.1407 | 0.4133 | 6.650 | 0.8980 | 0.8540 | 0.9157 |
|      | COMP | 0.1244 | 0.2975 | 6.649 | 0.9210 | 0.8490 | 0.9157 |
|      | MIXD | 0.1261 | 0.3382 | 6.649 | 0.9160 | 0.8500 | 0.9157 |
| COMP | RAND | 0.1416 | 0.4020 | 6.406 | 0.9000 | 0.9210 | 0.9053 |
|      | COMP | 0.1196 | 0.2858 | 6.405 | 0.9170 | 0.8910 | 0.9053 |
|      | MIXD | 0.1214 | 0.3234 | 6.405 | 0.9110 | 0.9040 | 0.9052 |
| MIXD | RAND | 0.1408 | 0.4154 | 6.406 | 0.8950 | 0.8900 | 0.9058 |
|      | COMP | 0.1197 | 0.2886 | 6.405 | 0.9150 | 0.8800 | 0.9058 |
|      | MIXD | 0.1212 | 0.3300 | 6.405 | 0.9100 | 0.8810 | 0.9059 |

Results using 1000 simulations as described in the text. The first column of the table gives data partition method for the covariance parameter estimation (COPE) using REML, which was one of random partitioning (RAND), compact partitioning (COMP), or a mix of compact with 10% randomly distributed (MIXD). The second column of the table uses covariance parameters as estimated in the first row, and gives the data partition method for fixed effects estimation (FEFE), which was one of RAND, COPE, or MIXD. RMSE, RMSPE, CI90, and PI90 are described in the text. RMSE$_1$ and RMSE$_2$ are for the first (spatially independent) and second (spatially patterned) covariates, respectively. Similarly, CI90$_1$ and CI90$_2$ are for first and second covariates, respectively.

**Table 4. Effect of partition sizes.**

| COPE | FEFE | RMSE$_1$ | RMSE$_2$ | RMSPE | CI90$_1$ | CI90$_2$ | PI90 | TIME$_C$ | TIME$_F$ |
|---|---|---|---|---|---|---|---|---|---|
| 25 | 25 | 0.147 | 0.645 | 6.77 | 0.938 | 0.845 | 0.932 | 2.821 | 3.328 |
| 25 | 50 | 0.131 | 0.340 | 6.77 | 0.955 | 0.807 | 0.932 | 2.821 | 1.249 |
| 25 | 100 | 0.133 | 0.372 | 6.77 | 0.930 | 0.833 | 0.932 | 2.821 | 0.758 |
| 25 | 200 | 0.130 | 0.346 | 6.77 | 0.938 | 0.810 | 0.932 | 2.821 | 0.730 |
| 50 | 25 | 0.146 | 0.593 | 6.14 | 0.943 | 0.963 | 0.909 | 3.031 | 3.328 |
| 50 | 50 | 0.121 | 0.290 | 6.13 | 0.897 | 0.900 | 0.909 | 3.031 | 1.249 |
| 50 | 100 | 0.122 | 0.309 | 6.13 | 0.912 | 0.922 | 0.908 | 3.031 | 0.758 |
| 50 | 200 | 0.120 | 0.288 | 6.13 | 0.917 | 0.922 | 0.909 | 3.031 | 0.730 |
| 100 | 25 | 0.143 | 0.634 | 6.13 | 0.930 | 0.882 | 0.906 | 4.802 | 3.328 |
| 100 | 50 | 0.121 | 0.304 | 6.13 | 0.900 | 0.885 | 0.907 | 4.802 | 1.249 |
| 100 | 100 | 0.122 | 0.322 | 6.13 | 0.905 | 0.917 | 0.906 | 4.802 | 0.758 |
| 100 | 200 | 0.120 | 0.299 | 6.13 | 0.910 | 0.910 | 0.906 | 4.802 | 0.730 |
| 200 | 25 | 0.144 | 0.637 | 6.13 | 0.927 | 0.877 | 0.905 | 12.760 | 3.328 |
| 200 | 50 | 0.121 | 0.300 | 6.13 | 0.897 | 0.887 | 0.905 | 12.760 | 1.249 |
| 200 | 100 | 0.122 | 0.322 | 6.13 | 0.905 | 0.905 | 0.905 | 12.760 | 0.758 |
| 200 | 200 | 0.120 | 0.300 | 6.13 | 0.907 | 0.902 | 0.905 | 12.760 | 0.730 |

Results are based on 1000 simulations, using the same simulation parameters as in Table 3. The first column of the table gives data partition sizes for the covariance parameter estimation (COPE), and the second column gives data partition size for fixed effects estimation (FEFE), while using covariance parameters as estimated in the first column. The columns RMSE$_1$, RMSE$_2$, RMSPE, CI90$_1$, CI90$_2$, and PI90 are the same as in Table 3. TIME$_C$ is the average time, in seconds, for covariance parameter estimation, and TIME$_F$ is the average time, in seconds, for fixed effects estimation.

partition size is the same for fixed effect and covariance parameter estimation, inverses of diagonal blocks can be passed from REML to fixed effects estimation, so another good strategy is to use block size 50 for both fixed effect and covariance parameter estimation.

## Variance estimation for fixed effects

In the section on estimating $\boldsymbol{\beta}$, we described three possible estimators for the covariance matrix of $\hat{\boldsymbol{\beta}}_{bd}$, with Eq (13) being theoretically correct, and faster alternatives Eqs (14) and (15). The alternative estimators will only be necessary for very large sample sizes, so to test their efficacy we simulated 1000 data sets with random sample sizes, from 10,000 to 100,000, using the SUMSINE method. We then fitted the covariance model, using compact partitions of size 50, and fixed effects, using partition sizes of 25, 50, 100, and 200. We computed the estimated covariance matrix of the fixed effects using Eqs (13)–(15), and evaluated performance based on 90% confidence interval coverage.

Results in Table 5 show that all three estimators, at all block sizes, have confidence interval coverage very close to the nominal 90% when estimating $\beta_1$, the independent covariate. However, when estimating the spatially-patterned covariate, $\beta_2$, the theoretical estimator has proper coverage for block sizes 50 and greater, while the two alternative estimators have proper coverage only for block size 50. It is surprising that the results for the alternative estimators are so specific to a particular block size, and these estimators warrant further research.

## Prediction with global estimate of $\boldsymbol{\beta}$

In the sections on point and block prediction, we described prediction using both a local estimator of $\boldsymbol{\beta}$, and the global estimator $\hat{\boldsymbol{\beta}}_{bd}$. To compare them, and examine the effect of the

**Table 5. CI90 for $\beta_1$ and $\beta_2$.**

| Part. Size | $\beta_1$ | | | $\beta_2$ | | |
|---|---|---|---|---|---|---|
| | Eq (13) | Eq (14) | Eq (15) | Eq (13) | Eq (14) | Eq (15) |
| 25 | 0.906 | 0.914 | 0.925 | 0.807 | 0.283 | 0.294 |
| 50 | 0.907 | 0.907 | 0.921 | 0.897 | 0.920 | 0.898 |
| 100 | 0.905 | 0.909 | 0.924 | 0.913 | 0.687 | 0.661 |
| 200 | 0.900 | 0.896 | 0.907 | 0.876 | 0.686 | 0.658 |

Results are based on 1000 simulations, using three different variance estimates, given by their equation numbers. Eq (13) is theoretically correct, while Eq (14) is based on empirical variation in $\hat{\boldsymbol{\beta}}$ among partitions, and Eq (15) is based on averaging the covariance matrices of $\hat{\boldsymbol{\beta}}$ among partitions.

**Table 6. Effect of number of nearest neighbors for RMSPE and PI90.**

| nNN | RMSPE$_1$ | RMSPE$_2$ | PI90$_1$ | PI90$_2$ | RMSPE$_3$ | PI90$_3$ | Time$_1$ | Time$_2$ | Time$_3$ |
|---|---|---|---|---|---|---|---|---|---|
| 25 | 6.62 | 6.36 | 0.908 | 0.907 | 0.204 | 0.912 | 0.6 | 2.4 | 6.9 |
| 50 | 6.45 | 6.33 | 0.907 | 0.907 | 0.201 | 0.907 | 1.2 | 3.0 | 7.5 |
| 100 | 6.37 | 6.32 | 0.907 | 0.907 | 0.201 | 0.904 | 4.4 | 6.3 | 10.5 |
| 200 | 6.34 | 6.31 | 0.907 | 0.907 | 0.200 | 0.905 | 23.9 | 25.7 | 29.0 |

Results are based on 1000 simulations, using the same simulation parameters as in Table 3. The first column of the table gives number of nearest neighbors. Time is average computing time in seconds. The subscript 1 indicates a local estimator of $\hat{\boldsymbol{\beta}}$ using Eq (6), while subscript 2 indicates global estimator of $\hat{\boldsymbol{\beta}}$ using Eq (17). The subscript 3 indicates the block predictor, Eq (21).

number of nearest neighbors, we simulated 1000 data sets as described in earlier, using compact partitions of size 50 for both covariance and fixed-effects estimation. We then predicted values on the gridded locations with 25, 50, 100, and 200 nearest neighbors.

Results in Table 6 show that prediction with the global estimator $\hat{\boldsymbol{\beta}}_{bd}$ had smaller RMSPE, especially with smaller numbers of nearest neighbors. As expected, predictors have lower RMSPE with more nearest neighbors, but gains are small after block size 50. Prediction intervals for both methods had proper coverage. The local estimator of $\boldsymbol{\beta}$ was faster because it used the local estimator of the covariance of $\boldsymbol{\beta}$, while predictions with $\hat{\boldsymbol{\beta}}_{bd}$ needed the global covariance estimator (Eq 13) to be used in Eq (19). Higher numbers of nearest neighbors took longer to compute, especially with numbers greater than 100. Of course, predictions for the block average had much smaller RMSPE than points. Again, prediction got better when using more nearest neighbors, but improvements were small with more than 50. Computing time for block averaging increased with number of neighbors, especially when greater than 100, and took longer than point predictions.

## A comparison of methods

To compare methods, we simulated 1000 data sets using GEOSTAT (partial sill was 10, range was 0.5 and nugget was 0.1) where we fix sample size at $n = 1000$, and the errors were standardized before adding fixed effects. We compared 3 methods: 1) estimation and prediction using the full covariance matrix for all 1000 points, 2) SPIN with compact blocks of 50 for both covariance and fixed effects parameter estimation, and 50 nearest-neighbors for prediction, and 3) nearest-neighbor Gaussian processes (NNGP). NNGP had good performance in [16] and software is readily available in the R package spNNGP [68]. For spNNGP, we used default

**Table 7. Comparison of 3 methods for fixed effects estimation and point prediction.**

| Method | RMSE$_1$ | RMSE$_2$ | RMSPE | CI90$_1$ | CI90$_2$ | PI90 | TIME |
|--------|----------|----------|-------|----------|----------|------|------|
| Full | 0.0088 | 0.0359 | 0.292 | 0.893 | 0.903 | 0.899 | 110.2 |
| SPIN | 0.0090 | 0.0380 | 0.292 | 0.908 | 0.913 | 0.906 | 3.0 |
| NNGP | 0.0090 | 0.0381 | 0.294 | 0.888 | 0.881 | 0.905 | 21.8 |

Data were simulated from 1000 random locations with a 40 × 40 prediction grid. The first column of the table gives the method, where Full uses the full 1000 × 1000 covariance matrix, SPIN uses spatial partitioning with compact blocks of size 50 and 50 nearest-neighbor prediction points. NNGP uses default parameters from R package for the conjugate prior method with a 25 × 25 search grid on phi and alpha. The columns RMSE$_1$, RMSE$_2$, RMSPE, CI90$_1$, CI90$_2$, and PI90 are the same as in Table 3. TIME is the average time, in seconds, for fixed effects estimation and prediction combined.

parameters for the conjugate prior method and a 25 × 25 search grid for phi and alpha, which were the dimensions of the search grid found in [16]. We stress that we do not claim this to be a definitive comparison among methods, as the developers of NNGP could surely make adjustments to improve performance. Likewise, partition size and number of nearest neighbors for prediction could be adjusted to optimize performance of SPIN for any given simulation or

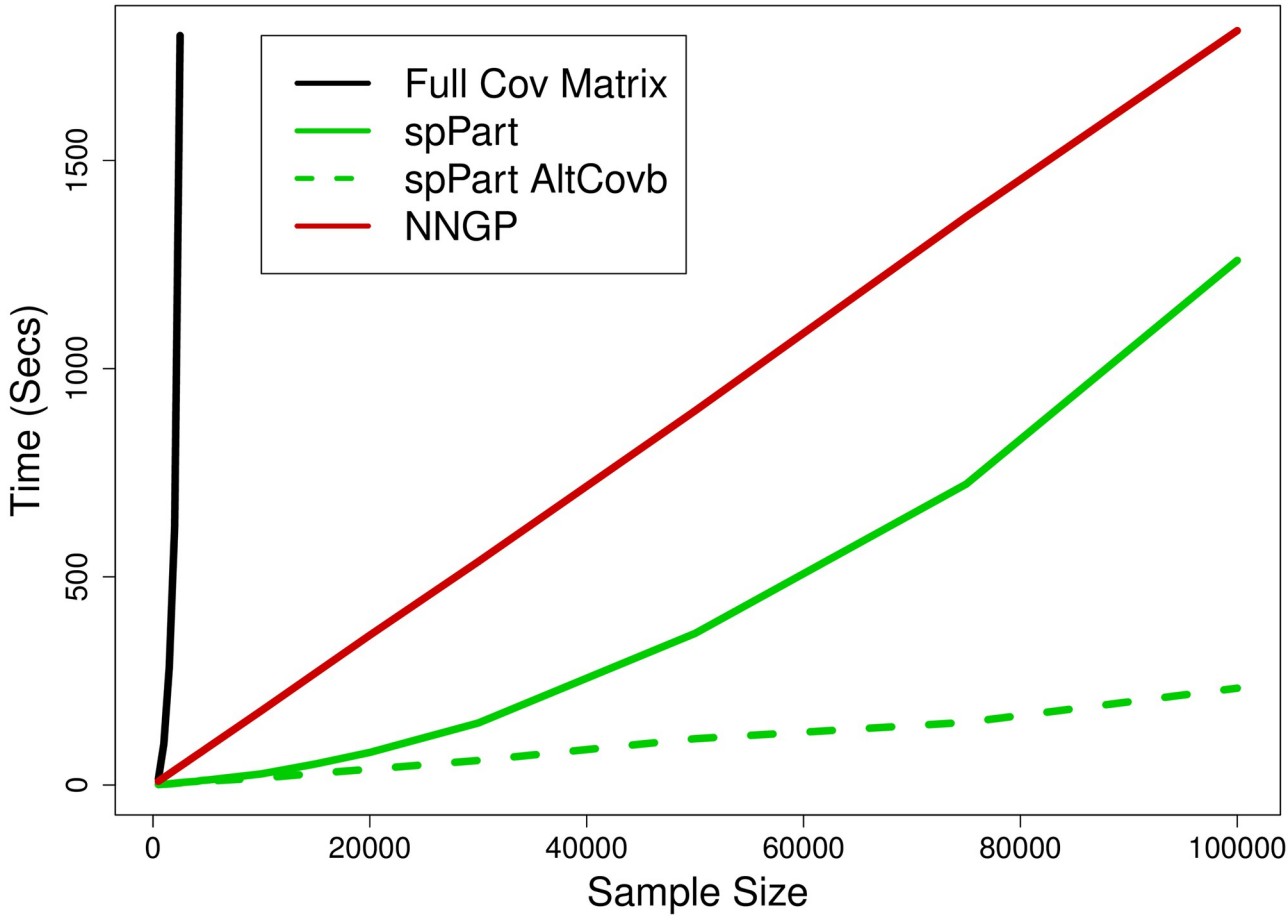

**Fig 4. Computing times as a function of sample size for three methods: 1) Full covariance matrix (black line), 2) NNGP (red line), and 3) SPIN (green lines).** For SPIN, the theoretically correct variance estimator (Eq 13) is solid green, while faster alternatives (Eqs 14 and 15) are dashed green.

data set. We offer these results to show that, broadly, SPIN and NNGP are comparable, and very fast, with little performance lost in comparison to using the full covariance matrix.

Table 7 shows that RMSE for estimation of the independent covariate, and the spatially-patterned covariate, were approximately equal for SPIN and NNGP, and only slightly worse than the full covariance matrix. RMSPE for SPIN was equal to the full covariance matrix, and both were just slightly better than NNGP. Confidence and prediction intervals for all three methods were very close to the nominal 90%.

Fig 4 shows computing times, using 5 replicate simulations, for each method for up to 100,000 records. Both NNGP and SPIN can use parallel processing, but here we used a single processor to remove any differences due to parallel implementations. Fitting the full covariance matrix with REML, which is iterative, took more than 30 minutes with sample sizes > 2500. Computing time for NNGP is clearly linear with sample size, while for SPIN, it is quadratic when using Eq (13), but linear when using the alternative variance estimators for fixed effects (Eqs 14 and 15). Using the alternative variance estimators, SPIN was about 10 times faster than NNGP, and even with quadratic growth when using Eq (13), SPIN was faster than NNGP for up to 100,000 records.

## Application to stream networks

We applied spatial indexing to covariance matrices constructed using stream network models as described for the motivating example in the Introduction. These are variance component models, with a tail-up component, a tail-down component, and a Euclidean-distance

**Table 8. Fixed effects table for Mid-Columbia river data.**

| Effect | $\hat{\beta}_{bd}$ | $se(\hat{\beta}_{bd})$ | $z$-value | Prob($> |z|$) |
|---|---|---|---|---|
| Intercept | 30.9324 | 5.8816 | 5.2592 | < 0.00001 |
| Elevation[1] | -4.0312 | 0.5052 | -7.9787 | < 0.00001 |
| Slope[2] | -0.1504 | 0.0289 | -5.2009 | < 0.00001 |
| Lakes[3] | 0.5287 | 0.1003 | 5.2690 | < 0.00001 |
| Precipitation[4] | -0.0018 | 0.0004 | -4.4639 | 0.00001 |
| Northing[5] | -0.6315 | 0.3002 | -2.1038 | 0.03565 |
| Flow[6] | -0.1118 | 0.0217 | -5.1429 | < 0.00001 |
| Drainage Area[7] | 0.0363 | 0.0236 | 1.5400 | 0.12388 |
| Canopy[8] | -0.0238 | 0.0033 | -7.1280 | < 0.00001 |
| Air Temperature[9] | 0.4538 | 0.0119 | 38.2106 | < 0.00001 |
| Discharge[10] | 0.0031 | 0.0140 | 0.2227 | < 0.82385 |

The $se(\hat{\beta}_{bd})$ is the standard error using Eq (13). The $z$-value is the estimate divided by its standard error. Prob($> |z|$) is the probability of getting the fixed effect estimate if it were truly 0, assuming a standard normal distribution.

[1] Elevation (m/1000) at sensor site

[2] Slope (100m/m) of stream reach of sensor site

[3] Percentage of watershed upstream of sensor site composed of lake or reservoir surfaces

[4] Mean annual precipitation (mm) in watershed upstream of sensor site

[5] Albers equal area northing coordinate (10km) of sensor site

[6] Percentage of the base flow to total flow of sensor site

[7] Drainage area (10,000 km$^2$) upstream of sensor site

[8] Riparian canopy coverage (%) of 1 km stream reach encompassing a sensor site

[9] Mean annual August air temperature (°C)

[10] Mean annual August discharge ($m^3$/sec)

component, each with 2 covariance parameters, along with a nugget effect; thus, there are 7 covariance parameters (4 partial sills, and 3 range parameters). A full covariance matrix was developed for these models [69], and we easily adapted it for spatial partitioning. We used compact blocks of size 50 for estimation, and 50 nearest neighbors for predictions. The 4 partial sill estimates were 1.76, 0.40, 2.57, and 0.66 for tail-up, tail-down, Euclidean-distance, and nugget effect, respectively. These indicate that tail-up and Euclidean-distance components dominated the structure of the overall autocovariance, and both had large range parameters. It took 7.98 minutes to fit the covariance parameters. The fitted fixed effects took an additional 2.15 minutes of computing time (Table 8), which are very similar to results found in [55]. Predictions for 65,099 locations are shown in Fig 5, which took 47 minutes.

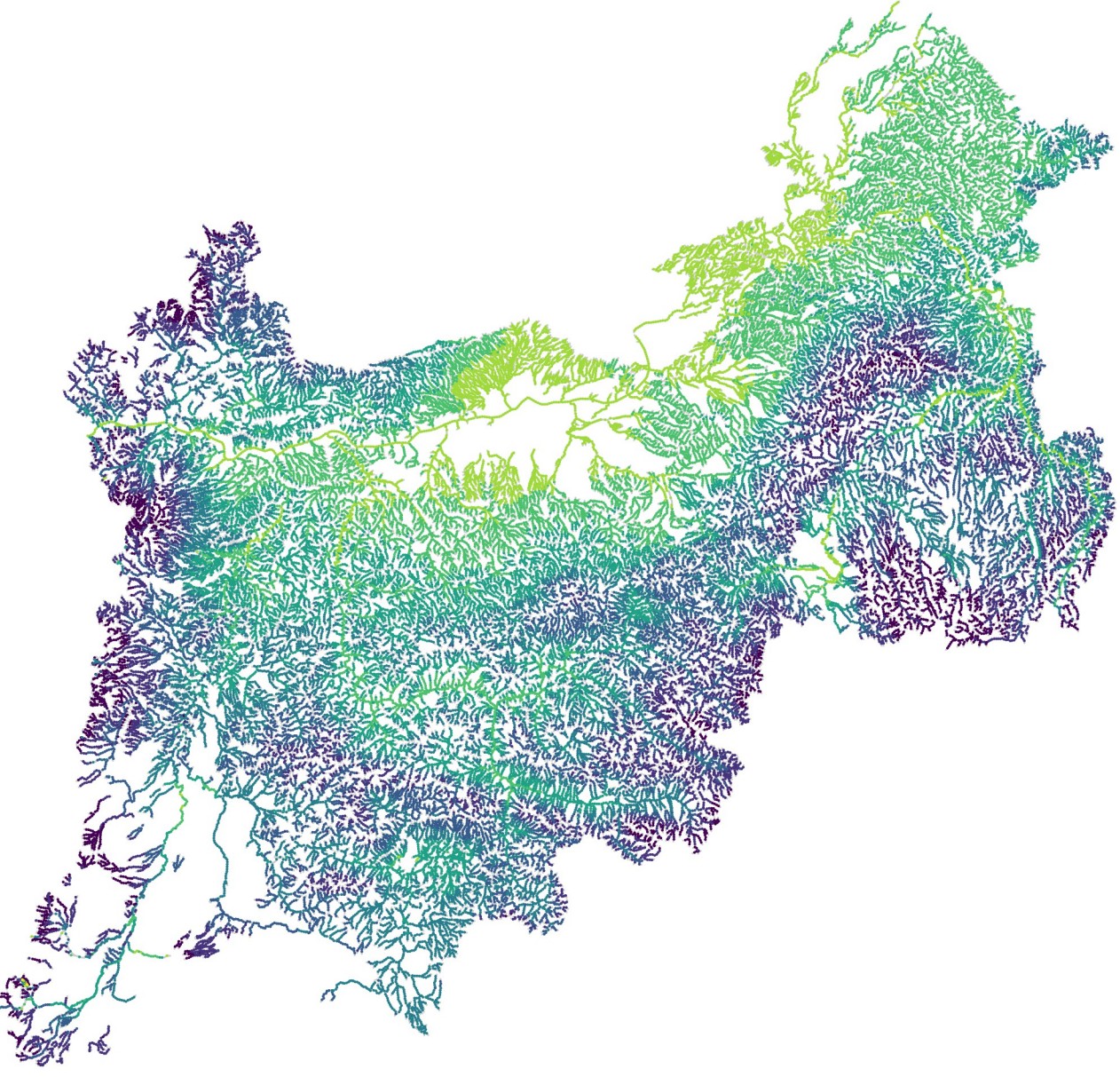

**Fig 5. Temperature predictions at 65,099 locations for the Mid-Columbia river.** Yellower colors are higher values, while bluer colors are lower values.

In summary, the original analysis [55] took 10 days of continuous computing time to fit the model and make predictions with a full $9521 \times 9521$ covariance matrix. Using SPIN, fitting the same model took about 10 minutes, with an additional 47 minutes for predictions. Note that these models take more time than Euclidean distance alone because there are 7 covariance parameters, and the tail-up and tail-down models use stream distance, which takes longer to compute. For this example, we used parallel processing with 8 cores when fitting covariance parameters and fixed effects, and making predictions, which made analyses considerably faster. We did not use block prediction, because that was not a particular goal for this study. However, it is generally important, and has been used for estimating fish abundance [70].

## Discussion and conclusions

We have explored spatial partitioning to speed computations for massive data sets. We have provided novel and theoretically correct development of variance estimators for all quantities. We proposed a globally coherent model for covariance and fixed effects estimation, and then use that model for improved predictions, even when those predictions are done locally based on nearest neighbors. We include block kriging in our development, which is absent among literature on big data for spatial methods.

Our simulations showed that, over a range of sample sizes, simulation methods, and range of autocorrelation, spatially compact partitions are best. There does not appear to be a need for "large blocks," as used in [54]. A good overall strategy, that combines speed without giving up much precision, is based on 50/50/50, where compact partitions of size 50 are used for both covariance parameter estimation and fixed effects estimation, and 50 nearest neighbors are used for prediction. This strategy compares very favorably with a default strategy for NNGP.

One benefit of the data indexing is that it extends easily to any geostatistical model with a valid covariance matrix. There is no need to approximate a Gaussian process. We provided one example for stream network models, but other examples include geometric anisotropy, nonstationary models, spatio-temporal models (including those that are nonseparable), etc. Any valid covariance matrix can be indexed and partitioned, offering both faster matrix inversions and parallel computing, while providing valid inferences with proper uncertainty assessment.

## Acknowledgments

We would like to thank Devin Johnson, Brett McClintock, Alan Pearse, and one anonymous reviewer for their reviews. The findings and conclusions in the paper are those of the author(s) and do not necessarily represent the views of the reviewers nor the EPA, BPA, and National Marine Fisheries Service, NOAA. Any use of trade, product, or firm names does not imply an endorsement by the US Government.

## Author Contributions

**Conceptualization:** Jay M. Ver Hoef.

**Data curation:** Erin E. Peterson, Daniel J. Isaak.

**Formal analysis:** Jay M. Ver Hoef, Michael Dumelle, Matt Higham.

**Investigation:** Jay M. Ver Hoef.

**Methodology:** Jay M. Ver Hoef, Michael Dumelle, Matt Higham, Erin E. Peterson, Daniel J. Isaak.

**Software:** Michael Dumelle, Matt Higham, Erin E. Peterson.

**Validation:** Jay M. Ver Hoef.

**Visualization:** Jay M. Ver Hoef.

**Writing – original draft:** Jay M. Ver Hoef.

**Writing – review & editing:** Jay M. Ver Hoef, Michael Dumelle, Matt Higham, Erin E. Peterson, Daniel J. Isaak.

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
