## [Decision Letter · Decision Letter 0]

17 May 2023

PONE-D-23-03446Indexing and Partitioning the Spatial Linear Model for Large Data SetsPLOS ONE

Dear Dr. Ver Hoef,

Thank you for submitting your manuscript to PLOS ONE. After careful consideration, we feel that it has merit but does not fully meet PLOS ONE’s publication criteria as it currently stands. Therefore, we invite you to submit a revised version of the manuscript that addresses the points raised during the review process.

We look forward to receiving your revised manuscript.

Kind regards,

Mohamed R. Abonazel, Ph.D.

Academic Editor

PLOS ONE

Journal Requirements:

"The project received financial support through Interagency Agreement DW-13-92434601-0 from the U.S. Environmental Protection Agency (EPA), and through Interagency Agreement 81603 from the Bonneville Power Administration (BPA), with the National Marine Fisheries Service, NOAA."

"JVH: The project received financial support through Interagency Agreement DW-13-92434601-0 from the U.S. Environmental Protection Agency (EPA), and through Interagency Agreement 81603 from the Bonneville Power Administration (BPA), with the National Marine Fisheries Service, NOAA. The funders had no role in study design, data collection and analysis, decision to publish, or preparation of the manuscript."

4. We note that Figures 1 and 5 in your submission contain [map/satellite] images which may be copyrighted. All PLOS content is published under the Creative Commons Attribution License (CC BY 4.0), which means that the manuscript, images, and Supporting Information files will be freely available online, and any third party is permitted to access, download, copy, distribute, and use these materials in any way, even commercially, with proper attribution. For these reasons, we cannot publish previously copyrighted maps or satellite images created using proprietary data, such as Google software (Google Maps, Street View, and Earth). For more information, see our copyright guidelines: http://journals.plos.org/plosone/s/licenses-and-copyright.

    a. You may seek permission from the original copyright holder of Figures 1 and 5 to publish the content specifically under the CC BY 4.0 license. 

Reviewers' comments:

Reviewer's Responses to Questions

**Comments to the Author**

1. Is the manuscript technically sound, and do the data support the conclusions?

Reviewer #1: Yes

2. Has the statistical analysis been performed appropriately and rigorously? 

Reviewer #1: Yes

3. Have the authors made all data underlying the findings in their manuscript fully available?

Reviewer #1: Yes

4. Is the manuscript presented in an intelligible fashion and written in standard English?

Reviewer #1: No

5. Review Comments to the Author

Reviewer #1: Indexing and Partitioning the Spatial Linear Model for Large Data Sets

Few major comments are attached for the improvement of the article.

1. There are too many variables (parameters) in this article, I suggest to add a table of acronym to define each variable separately.

2. What is Eq (1)? You should write Equation (1).

3. Σ is the standard notation for sum, but you used for different concept. You should avoid using standard notation for different concepts.

4. In the literature review, you used 13-16, 17-19, 20-22, 27-29. You should explain these separately.

5. Figure 1 is not on the place of Figure 1, you should place your figures over there where you claimed.

6. Where did you get the methodology? Is it standard or your own novelty? You did not cited a single reference there.

7. There are too many short forms in this article. I suggest again to create a new table to define each short form separately. See Page 12/26.

8. Tables are not on the place of tables, you should place your tables over there where you claimed.

9. The literature about Partitioning is very limited, I recommend to add more detail about graph theory. I recommend to add the following basics, Notes on the Localization of Generalized Hexagonal Cellular Networks, Mathematics. DOI: 10.3390/math11040844. Verification of some topological indices of Y-junction based nanostructures by M-polynomials. Journal of Mathematics. DOI: 10.1155/2022/8238651. Sharp bounds on partition dimension of hexagonal Mobius ladder. Journal of King Saud University-Science, Dec. 2021. DOI:10.1016/j.jksus.2021.101779. Metric-based resolvability of polycyclic aromatic hydrocarbons. European Physical Journal Plus. DOI:10.1140/epjp/s13360-021-01399-8

10. In page 17/26, line 496, what is covariance matrice? A typo.

11. There are many typos in this article, you should recheck this article.

12. References are not in the same pattern see 59 and 60 (page section).

13. See the difference in journal name of ref 62 with others.

14. Figures are very blur unable to understand, kindly revise the quality of pictures.

15. Try to write the article in understandable way, you are teaching newcomers not only publishing articles.

16. Elaborate each section in an understandable way, make section and enumerate them.

6. PLOS authors have the option to publish the peer review history of their article (what does this mean?). If published, this will include your full peer review and any attached files.

Reviewer #1: No

---

## [Author Response · Author response to Decision Letter 0]

26 Jun 2023

Please see the attached response letter to the editor and reviewer. Thank you very much for your reviews.

---

## [Decision Letter · Decision Letter 1]

2 Aug 2023

PONE-D-23-03446R1Indexing and Partitioning the Spatial Linear Model for Large Data SetsPLOS ONE

Dear Dr. Ver Hoef,

Thank you for submitting your manuscript to PLOS ONE. After careful consideration, we feel that it has merit but does not fully meet PLOS ONE’s publication criteria as it currently stands. Therefore, we invite you to submit a revised version of the manuscript that addresses the points raised during the review process.

We look forward to receiving your revised manuscript.

Kind regards,

Mohamed R. Abonazel, Ph.D.

Academic Editor

PLOS ONE

Journal Requirements:

Additional Editor Comments:

The authors are requested to make appropriate modifications to this manuscript as suggested by the reviewer.

Reviewers' comments:

Reviewer's Responses to Questions

**Comments to the Author**

1. If the authors have adequately addressed your comments raised in a previous round of review and you feel that this manuscript is now acceptable for publication, you may indicate that here to bypass the “Comments to the Author” section, enter your conflict of interest statement in the “Confidential to Editor” section, and submit your "Accept" recommendation.

Reviewer #1: All comments have been addressed

Reviewer #2: All comments have been addressed

2. Is the manuscript technically sound, and do the data support the conclusions?

Reviewer #1: Partly

Reviewer #2: Yes

3. Has the statistical analysis been performed appropriately and rigorously? 

Reviewer #1: No

Reviewer #2: Yes

4. Have the authors made all data underlying the findings in their manuscript fully available?

Reviewer #1: No

Reviewer #2: Yes

5. Is the manuscript presented in an intelligible fashion and written in standard English?

Reviewer #1: Yes

Reviewer #2: Yes

6. Review Comments to the Author

Reviewer #1: You may accept this article from my side. Authors addressed all the suggested comments from my side. It is quite to accept this article.

Thank you

Reviewer #2: The authors present an approximate inference method for large spatial datasets by indexing and partitioning the data into blocks and taking these blocks to be independent, thus reducing the computational load. While this approach is not new, the authors present enough variations and innovation to differentiate it from the already published methods. The authors provide simulated examples where the fixed effects in the model were estimated with fair accuracy. The method is applied to a real dataset with success.

The article is written well and is easily readable for people who engage in practicing spatial statistical methods. However, I have the following comments about the work:

1. Is the primary objective of this method to efficiently estimate the covariate effects? You do not present covariance parameter estimation results anywhere.

2. I am skeptical about the spatial prediction performance of the method. As per the described data generation method, the magnitude of the response variable Y would be around 40 at most. From Tables 3 and 4, the RMSPE is of similar magnitude. This indicates rather poor performance and since you do not compare these with any other method, it is hard to say if the method is working well or not. Please correct me about the magnitude of Y, if I am wrong.

3. The RMSPE in Table 7 is surprisingly low compared to the numbers in Tables 3, 4 and 6. Are the data generated similarly, with same scale of Y in both cases? If so, what lead to such huge improvements?

4. This is a minor comment, but the spatial range parameter being bigger than 1 (or even 0.5) for a field of size unit square is unrealistic. But for investigating purposes, it is fine.

5. Eq (19) has a typo, it should be Vj in the second term and not Vi

7. PLOS authors have the option to publish the peer review history of their article (what does this mean?). If published, this will include your full peer review and any attached files.

Reviewer #1: No

Reviewer #2: No

---

## [Author Response · Author response to Decision Letter 1]

24 Aug 2023

Please see the rebuttal letter. Thank you.

---

## [Decision Letter · Decision Letter 2]

11 Sep 2023

Indexing and Partitioning the Spatial Linear Model for Large Data Sets

PONE-D-23-03446R2

Dear Dr. Ver Hoef,

We’re pleased to inform you that your manuscript has been judged scientifically suitable for publication and will be formally accepted for publication once it meets all outstanding technical requirements.

Kind regards,

Mohamed R. Abonazel, Ph.D.

Academic Editor

PLOS ONE

Additional Editor Comments (optional):

Reviewers' comments:

Reviewer's Responses to Questions

**Comments to the Author**

1. If the authors have adequately addressed your comments raised in a previous round of review and you feel that this manuscript is now acceptable for publication, you may indicate that here to bypass the “Comments to the Author” section, enter your conflict of interest statement in the “Confidential to Editor” section, and submit your "Accept" recommendation.

Reviewer #1: All comments have been addressed

Reviewer #2: All comments have been addressed

2. Is the manuscript technically sound, and do the data support the conclusions?

Reviewer #1: Yes

Reviewer #2: Yes

3. Has the statistical analysis been performed appropriately and rigorously? 

Reviewer #1: Yes

Reviewer #2: Yes

4. Have the authors made all data underlying the findings in their manuscript fully available?

Reviewer #1: Yes

Reviewer #2: Yes

5. Is the manuscript presented in an intelligible fashion and written in standard English?

Reviewer #1: Yes

Reviewer #2: Yes

6. Review Comments to the Author

Reviewer #1: You may accept this article from my side. Authors addressed all the suggested comments from my side. It

is quite to accept this article.

Reviewer #2: The authors have addressed all my concerns. Great work on the article! With the revised tables, the work really shows that it is a good approximate inference method for large spatial dataset

7. PLOS authors have the option to publish the peer review history of their article (what does this mean?). If published, this will include your full peer review and any attached files.

Reviewer #1: **Yes: **Muhammad Azeem

Reviewer #2: No

---

## [Editor Report · Acceptance letter]

23 Oct 2023

PONE-D-23-03446R2 

Indexing and partitioning the spatial linear model for large data sets 

Dear Dr. Ver Hoef:

I'm pleased to inform you that your manuscript has been deemed suitable for publication in PLOS ONE. Congratulations! Your manuscript is now with our production department. 

Kind regards, 

on behalf of

Dr Mohamed R. Abonazel 

Academic Editor

PLOS ONE